# Learning Spatially-Adaptive Squeeze-Excitation Networks for Image Synthesis and Image Recognition

## Abstract

Learning light-weight yet expressive deep networks in both image synthesis and image recognition remains a challenging problem. Inspired by a more recent observation that it is the data-specificity that makes the multi-head self-attention (MHSA) in the Transformer model so powerful, this paper proposes to extend the widely adopted light-weight Squeeze-Excitation (SE) module to be spatially-adaptive to reinforce its data specificity, as a convolutional alternative of the MHSA, while retaining the efficiency of SE and the inductive basis of convolution. It presents two designs of spatially-adaptive squeeze-excitation (SASE) modules for image synthesis and image recognition respectively. For image synthesis tasks, the proposed SASE is tested in both low-shot and one-shot learning tasks. It shows better performance than prior arts. For image recognition tasks, the proposed SASE is used as a drop-in replacement for convolution layers in ResNets and achieves much better accuracy than the vanilla ResNets, and slightly better than the MHSA counterparts such as the Swin-Transformer and Pyramid-Transformer in the ImageNet-1000 dataset, with significantly smaller models.

## 1   Introduction

Both image synthesis and image recognition remain challenging problems in computer vision and machine learning. Despite remarkable progress has been made since the recent resurgence of deep neural networks (DNNs), both synthesizing high-fidelity and high-resolution images and classifying images accurately at scale typically entails computationally expensive training and inference, which have shown to lead to potential environmental issues due to the carbon footprint [1]. Along with the progress, learning light-weight yet highly-expressive deep models also remains an important and interesting research direction, especially with less data. This paper focuses on learning low-shot (e.g., 100 to 1000 images in training) and one-shot image synthesis models and on learning smaller yet expressive models for image recognition at scale.

Consider generative adversarial networks (GANs), state-of-the-art methods such as BigGANs [2] and StyleGANs [3, 4] utilize ResNets as their backbones. Although powerful, as the resolution of synthesized images goes higher, the width and the depth of a generator network goes wider and deeper accordingly, leading to much increased memory footprint and longer training time. The more recent Transformer based models often further increase the complexities [5]. To address these issues, Liu et al [6] present a FastGAN approach which introduces a Skip-Layer channel-wise Excitation (SLE) module to reduce the computation and memory complexities of both generator and discriminator networks (Fig. 1), together with exploiting the differentiable data augmentation methods [7]. FastGANs have shown exciting results which outperform the well-known and powerful StyleGANv2 [4] under the low-shot training settings.

Submitted to 36th Conference on Neural Information Processing Systems (NeurIPS 2022). Do not distribute.

What make the light-weight SE/SLE an effective drop-in module? One possible explanation lies in its data specificity that enables on-the-fly feature modulation between feature responses in both training and inference. More recently, the data specificity of the multi-head self-attention (MHSA) module in the Transformer model has been shown to be the key to its representational power (rather than its long-range contextual modeling capability) [10]. However, SE/SLE is a channel-wise realization of the data specificity, without accounting for the spatial feature modulation/attention (the spatial dimensions are entirely squeezed). So, **a question naturally arises:** *Can we extend SE/SLE to be spatially-adaptive, such that we can build a light-weight convolutional alternative to the MHSA to retain the efficiency of SE/SLE and the inductive basis of convolution for both fast and low-shot image synthesis and large scale image recognition applications.*

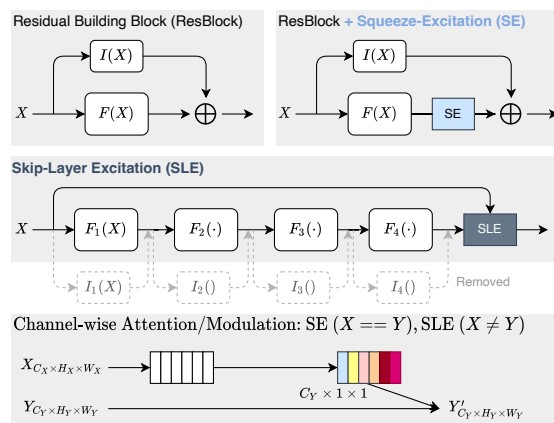

Figure 1: *Top*: Illustration of the ubiquitous residual building block [8], and its reinforced variant with the popular squeeze-excitation (SE) module [9] that learns channel-wise feature attention. $F(X)$ represents the transformation applied to an input feature map $X$. $I(X)$ represents the skip connection. *Middle:* Illustration of the Skip-Layer Excitation (SLE) module in FastGANs [6]. With the SLE, the original layer-wise skip-connections are removed to reduce computational and memory complexities (e.g., $I_1()$ to $I_4()$ shown in dotted blocks). *Bottom*: Both SE and SLE realize channel-wise feature attention/modulation to re-calibrate the feature maps.

To address the question, this paper proposes to learn spatially-adaptive squeeze-excitation (SASE) networks with two realizations for image synthesis and image recognition respectively (Fig. 2). We give a brief overview of the proposed modules below.

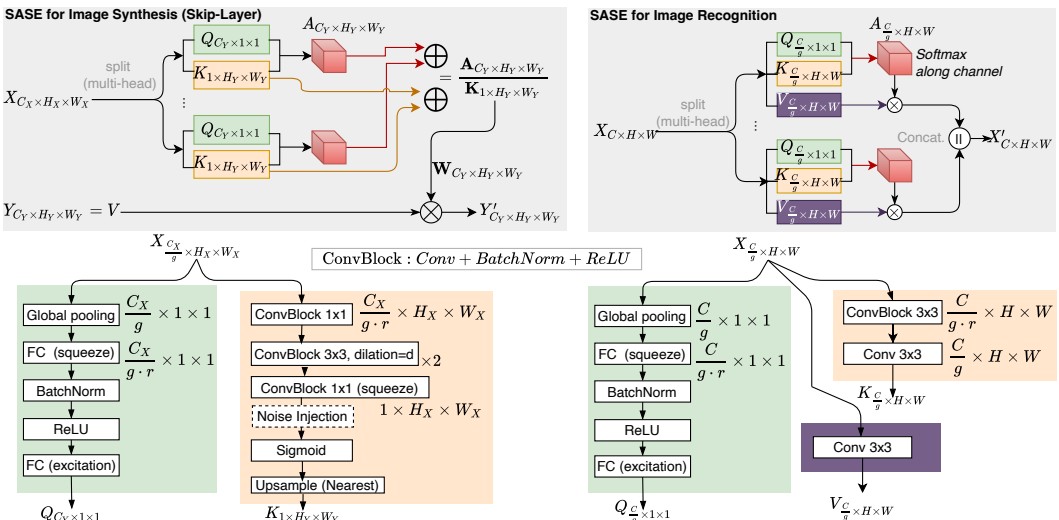

Figure 2: Illustration of the proposed SASE module. It resembles the multi-head computation in the Transformer model [11]. It exploits different strategies in computing the attention. $g$ represents the number of heads/groups used to split the input along the channel dimension (e.g., $g = 4$), and $r$ represents the squeezing ratio (e.g., $r = 4$). See text for details.

**SASE for Image Synthesis.** The left of Fig. 2 illustrates the proposed design of SASE to facilitate efficient low-shot and one-shot image synthesis. Unlike the channel-wise 1-D attention weights for a target feature map $X$ in both the SE and SLE modules (i.e., $C_Y \times 1 \times 1$, see the bottom of Fig. 1), the proposed SASE aims to learn a full 3D attention weights (i.e., $A_{C_Y \times H_Y \times W_Y}$) in a multi-head way. Each 3D attention matrix is computed by broadcasting and multiplying the learned Query vector $Q_{C_Y \times 1 \times 1}$ (accounting for the latent style information for image synthesis by squeezing the spatial dimensions) and the learned Key map $K_{1 \times H_Y \times W_Y}$ (accounting for the spatial mask by squeezing the channel dimensions). The multi-head 3D attention matrices are summed together and normalized

by the sum of the Key maps. The resulting final 3D attention matrix used for modulating the feature map $Y_{C_Y \times H_Y \times W_Y}$ integrates both spatial and channel-wise attention. This 3D attention matrix enables richer information flow from a source feature map $X$ to a target one $Y$, which leads to better generation quality for low-shot and one-shot image synthesis. Fig. 3 shows the deployment of the SASE module in FastGANs [6] and SinGANs [12].

**SASE for Image Recognition.** As illustrated in the right of Fig. 2, the learned full 3D attention matrix resembles the role of the self-attention weights in the Transformer model [11]. In the Transformer model, the attention is explicitly calculated between all pairs of "tokens" (e.g., patches after embedding) after the query and key transformation respectively. The resulting full attention matrix is thus quadratic in terms of the number of "tokens". In the proposed SASE, the channel-wise attention squeezes all spatial locations, and the resulting 1D Query (style) vector conveys/squeezes information from all locations. The spatial Key map squeezes the channels, and the resulting 2D spatial masks (heatmaps) forms the soft grouping of pixels in each mask. The resulting 3D attention matrix thus implicitly measure the attention weights used in the Transformer model at a much coarser level, but will be more efficient to compute. Softmax is applied along the channel dimension of the 3D attention matrix. It is then used to re-calibrate the output of the Value (convolutional response) map in an element-wise / spatially-adaptive way. As shown in Fig. 4, it is used to replace all the 3x3 convolution layers in a feature backbone (e.g., ResNet-50). It achieves much better accuracy with significantly smaller model complexity in ImageNet-1000 [13].

**Our Contributions.** In summary, this paper makes three main contributions as follows: (i) It presents a Spatially-Adaptive Squeeze-Excitation (SASE) module with two realizations for better learning of generative models from low-shot / one-shot images, and for large-scale discriminative learning like the Transformer model, but in a more efficient way, respectively. (ii) It shows significantly better performance for high-resolution image synthesis at the resolution of $1024 \times 1024$ when deployed in the FastGANs [6], while retaining the efficiency. It also shows better performance in image classification and object detection with much smaller models. (iii) It enables a simplified workflow for SinGANs [12], and shows a stronger capability of preserving image structures than prior arts.

## 2 Approach

### 2.1 The SASE for Image Synthesis

**Uncondiational Image Synthesis.** The goal is to learn a generator network which maps a latent code to an image,

$$x = G(z; \Theta_G), \tag{1}$$

where $z$ represents a 1-D latent code (e.g., in FastGANs [6] or a 2-D latent code (e.g., in Sin-GANs [12]), which is typically drawn from standard Normal distribution (i.e., white noise). $\Theta_G$ collects all the parameters of the generator network $G$. Given a latent code, it is straightforward to generate an image.

**FastGANs.** As shown in the left of Fig. 3, the generator network used in FastGANs [6] adopts a minimally-simple yet elegantly chosen design methodology. Given an input latent code, the initial block applies the transpose convolution to map the latent code to a $4 \times 4$ feature map. Then, a composite block (UpCompBlock) and a plain block (UpBlock) are interleaved to map the $4 \times 4$ feature map to the one at a given target resolution (e.g., $1024 \times 1024$). Batch normalization [14] and gated linear unit (GLU) [15] are used in the building blocks. In a composite upsample block, noise injection is used right after the convolution operation. Please refer the original paper [6] for details.

**SinGANs.** The right-top of Fig. 3 shows a stage in the generator of SinGANs [12]. A SinGAN is progressively trained from a chosen low resolution to the resolution of the single input image. Following the notation usage in SinGANs [12], the start resolution is indexed by $N$ and the end resolution by 0. At the very beginning, a 2-D latent code, $z_N$ is sampled and the initial generator $X'_N = G_N(z_N; \theta_N)$ is trained under the vanilla GAN settings. Then, at the stage $n$ ($N > n \geq 0$), the generator $G_n$ has been progressively grown from $G_N$, and we have $X'_n = G_n(z_n, (X'_{n+1}) \uparrow; \theta_n)$, where $(X'_{n+1}) \uparrow$ represents the up-sampled synthesized image from the previous stage $n + 1$ with respect to the predefined ratio used in the construction of the image pyramid. More specifically, $X'_n = (X'_{n+1}) \uparrow + \psi_n(z_n + (X'_{n+1}) \uparrow; \theta_n)$. More details are referred to the original paper [12].

With the proposed SASE module, we substantially change the workflow of the generator as shown in the right-bottom of Fig. 3: Each stage of the vanilla SinGAN is to learn the residual image on

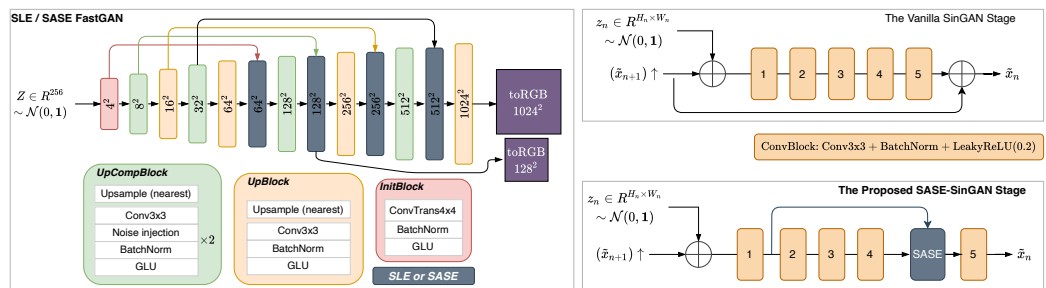

Figure 3: *Left:* The generator network of FastGANs [6] and the drop-in replacement of the SLE module by our SASE module. The network specification is reproduced based on the officially released code of FastGANs (link). *Right:* Illustration of deploying the proposed SASE module in SinGANs [12]. See text for details.

top of the output from the previous stage. As we shall elaborate, the proposed SASE module is spatially-adaptive in modulating a target feature map using a source feature map, so we remove the residual learning setting. We keep the discriminator of SinGANs unchanged in our experiments.

**The SASE Module.** Fig. 2 shows the proposed SASE module. We first compare the formulation between the SE module [9], the SLE module [6] and the proposed SASE module. Focusing on how an input target feature map $Y$ is transformed to the output feature map $Y'$, denote by $Y = (\mathbf{y}_1, \cdots, \mathbf{y}_{\mathbf{C_Y}})$ where $\mathbf{y}_c$ represents a single channel slice of the tensor $Y$ for $1 \leq c \leq C_Y$, we have,

$$\text{SE: } Y' = (\alpha_1 \cdot \mathbf{y}_1, \cdots, \alpha_C \cdot \mathbf{y}_{C_Y}), \tag{2}$$

$$\text{SLE: } Y' = (\beta_1 \cdot \mathbf{y}_1, \cdots, \beta_C \cdot \mathbf{y}_{C_Y}), \tag{3}$$

where the channel importance coefficient $\alpha_c = \mathcal{F}_{SE}(Y)$ in the SE module, and $\beta_c = \mathcal{F}_{SLE}(X)$ in the SLE module. So, the SE module realizes self-attention between channels (a.k.a. "neurons"), while the SLE module realizes cross-attention. And, the former is a special case of the latter when $X = Y$. Both $\alpha_c$ and $\beta_c$ are scalar and shared by all spatial locations in the same channel slice. For discriminative learning tasks such as image classification, this channel-wise feature attention works very well since spatial locations will be discarded by the classification head sub-network (typically via a global average pooling followed by a fully-connected layer). For image synthesis tasks whose outputs are location-sensitive, it may not be sufficient to deliver the entailed modulation effects.

The proposed SASE module aims to facilitate spatially-adaptive attention by extending the SLE module. It learns a 3D weight matrix $\mathbf{W}_{C_Y \times H_Y \times W_Y}$ from the source feature map $X$ in modulating the target feature map $Y_{C_Y \times H_Y \times W_Y}$ (that is to "pay full attention"), and we have,

$$\text{SASE: } Y' = \mathbf{W} \circ Y, \tag{4}$$

where $\circ$ represent the Hadamard product.

**Learning the spatially-adaptive attention matrix W from $X$.** We want to distill two types of information: One represents 1D latent style codes (as the Query vector) that are informed by the source feature map $X$, and then are used to induce the modulated target feature map $Y'$ to focus on. The other reflects 2D latent spatial masks (as the Key maps) that are used to distribute the latent Query/style codes. Decoupling these two is beneficial to enable them learning faster and more accurate.

**Decoupling the Style and Layout.** We decouple the channels ("neurons") in an input source feature map by splitting them into a number of groups (e.g., 4), that is to exploit mixture modeling or clustering of the "neurons" in a building block, as suggested by the theoretical study of how to construct an optimal neural architecture in a layer-wise manner with a set of constraints satisfied [16] and as typically done in the MHSA of the Transformer model. For each group, we apply the decoupled channel-wise and spatial transformation for learning the latent style codes and the latent spatial masks concurrently.

To sum up, from the channel-wise attention branches, we compute a group of $g$ latent Query/style vectors, $Q_{C_Y \times 1 \times 1}$'s. From the spatial attention branches, we compute a group of $g$ latent spatial Key masks, $K_{1 \times H_Y \times W_Y}$. Then, the 3-D weight matrix $\mathbf{W}$ in Eqn. 4 is computed by,

$$\mathbf{W} = \frac{\sum_{i=1}^{g}(\mathbb{Q}^i \circ \mathbb{K}^i)}{\sum_{i=1}^{g} \mathbb{K}^i}, \tag{5}$$

where $\mathbb{Q}$ and $\mathbb{K}$ are broadcasted from $Q$ and $K$ to match the dimensions respectively.

## 2.2 The SASE for Image Recognition

The right of Fig. 2 illustrates the proposed SASE for image recognition. Its implementation is straightforward following the discussions above. It is used for substituting the 3x3 Convolutions in a network (e.g., the ResNets [8]), as shown in the right of Fig. 4.

More specifically, we can compare the computation workflows between our SASE and the MHSA module. Let $c$ be the input dimension (i.e., $c = \frac{C}{r}$), $g$ the number of heads, and $d = \frac{c}{g}$ the head dimension. For simplicity, we omit the additive positional encoding used in integrating the MHSA in ResNets. Please refer to [17] for more details. Following the terminology used in the Transformer model [11], denote by $N = H \times W$ the number of "tokens". In MHSA and SASE, the query, key and value are then defined respectively by:

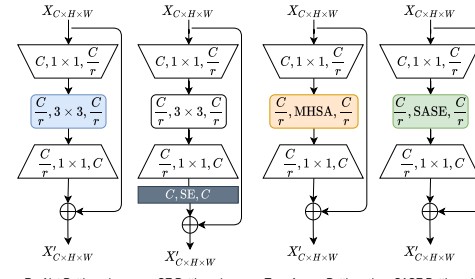

Figure 4: Comparisons between two variants of the vanilla ResNet bottleneck block [8] (left) with the proposed SASE bottleneck: the SE bottleneck [9] is a widely adopted design, and the Transformer bottleneck is a recently proposed method [17]. $r$ is the bottleneck ratio (e.g., $r = 4$).

$$\text{MHSA:} \quad Z_{g \times N \times d} = \text{Reshape}(W^Z_{c \times c} \times X_{c \times H \times W}), \quad Z \in \{Q, K, V\} \tag{6}$$

$$A^i_{N \times N} = \text{Softmax}((Q^i_{N \times d} \cdot K^i_{d \times N})/\sqrt{d}), \quad i = 1, \cdots g, \tag{7}$$

$$X'_{c \times H \times W} = \text{Reshape}(\text{Concat}(A^i_{N \times N} \times V^i_{N \times d})^g_{i=1}); \tag{8}$$

$$\text{SASE:} \quad Q^i_{d \times 1 \times 1} = \text{SE}(X^i_{d \times H \times W}), \quad i = 1, \cdots g, \tag{9}$$

$$K^i_{d \times H \times W} = \text{Conv3x3}(\text{Conv3x3BNReLU}(X^i_{d \times H \times W})), \tag{10}$$

$$V^i_{d \times H \times W} = \text{Conv3x3}(X^i_{d \times H \times W}), \tag{11}$$

$$A^i_{d \times H \times W} = \text{Softmax}(Q^i_{d \times 1 \times 1} \circ K^i_{d \times H \times W}), \tag{12}$$

$$X'_{c \times H \times W} = \text{Concat}(A^i_{d \times H \times W} \circ V^i_{d \times H \times W})^g_{i=1}, \tag{13}$$

where the MHSA often suffers from the quadratic complexities of computint time and memory footprint in terms of the input number of "tokens" ($N$). Our SASE can retain linear complexities.

In terms of maintaining the on-the-fly data-specificity as pointed out in [10], our SASE offers an alternative and efficient computation workflow. In our SASE, the query attempts to summarize information from all spatial locations, and the key attempts to maintain the locality. The resulting attention via broadcasting and multiplying the query and key facilitate integrating the global and local information, which is then used to modulate the value.

## 3 Experiments

In this section, we test the proposed SASE module on four tasks: low-shot image synthesis using FastGANs [6], one-shot image synthesis using SinGANs [12], ImageNet-1000 classification using ResNets [8], and MS-COCO object detection and instance segmentation using Mask R-CNN [18] with ResNets backbone. **Our PyTorch source code is provided in the supplementary materials**.

### 3.1 Low-Shot Image Synthesis Results

**Data and Settings.** We adopt the datasets used in the vanilla FastGANs [6] for fair comparisons with the SLE. There are 5 categories tested at the resolution of $256 \times 256$ each of which uses around 100 training images. There are 7 categories tested at the resolution of $1024 \times 1024$, four of which use around 1000 training images and the remaining of which use around 100 training images. The categories are listed in Table 1. We follow settings used in the official code of FastGANs. One thing worth clarifying is the output size of the discriminator. There are two different settings used for different categories in the original experiments by FastGANs [6]: $1 \times 1$ or $5 \times 5$, which show different performance on different categories. For simplicity, we use $5 \times 5$ consistently throughout the experiments as the output size, so some of the results of the proposed SASE module could be further improved. A single GPU is used in training.

**Metrics.** To evaluate the quality of synthesized images, we adopt the widely used Fréchet Inception Distance (FID) [19] and Kernel Inception Distance (KID) [20] metrics. KID has better sample-efficiency and lower estimation bias than FID, more suitable for low-shot image synthesis. We further use the density and coverage [21] metric for evaluating the reliable fidelity and diversity, where we

use the default $k$-nearest neighbours with $k = 5$. To assess the potential memorization in low-shot image synthesis methods, we use the Kolmogorov-Smirnov (KS) $p$-value proposed in the latent recovery method [22]. *We use $\uparrow$ and $\downarrow$ alongside each of the metric in the tables to indicate whether the larger/smaller its value is, the better a model is.*

| Metric | Method (**DiffAug**) | 256×256, ~100 images per category | | | | | 1024× 1024, ~1000 images | | | | 1024× 1024, ~100 images | | |
|---|---|---|---|---|---|---|---|---|---|---|---|---|---|
| | | Obama | Dog | Cat | Grumpy Cat | Panda | FFHQ | Art | Flower | Pokemon | AnimeFace | Skulls | Shells |
| FID↓ | SLE | 41.05 | 50.66 | 35.11 | 26.65 | 10.03 | 44.3 | 45.08 | 31.7 | 57.19 | 59.38 | 130.05 | 155.47 |
| | SPAP | 51.98 | 58.46 | 54.31 | 30.15 | 14.41 | 78.37 | 61.89 | 60.15 | 114.98 | 93.53 | 118.09 | 160.12 |
| | CBAM | 40.05 | 52.35 | 36.14 | 26.89 | 10.14 | 58.23 | 58.12 | 44.13 | 76.76 | 84.45 | 125.61 | 156.76 |
| | **SASE (ours)** | **36.4** | **49.99** | **33.55** | **26.01** | **9.48** | **39.59** | **43.46** | **29.90** | **51.2** | **54.22** | **101.16** | **140.45** |
| KID↓ | SLE | 0.012 | 0.014 | 0.006 | 0.007 | 0.004 | 0.012 | 0.011 | **0.006** | 0.014 | 0.018 | 0.054 | 0.068 |
| | SPAP | 0.045 | 0.026 | 0.014 | 0.013 | 0.009 | 0.21 | 0.54 | 0.019 | 0.11 | 0.15 | 0.045 | 0.11 |
| | CBAM | 0.012 | 0.016 | 0.007 | 0.007 | 0.004 | 0.15 | 0.45 | 0.012 | 0.058 | 0.13 | 0.051 | 0.071 |
| | **SASE (ours)** | **0.005** | **0.012** | **0.004** | **0.004** | **0.002** | **0.011** | **0.009** | **0.006** | **0.011** | **0.014** | **0.030** | **0.044** |
| Density↑ | SLE | 1.31 | 0.79 | 0.95 | 1.25 | 1.78 | 1.18 | 1.38 | 0.85 | 1.14 | 1.17 | 0.90 | 0.29 |
| | SPAP | 0.91 | 0.53 | 0.89 | 1.01 | 1.32 | 0.81 | 0.74 | 0.66 | 0.54 | 0.61 | 0.92 | 0.27 |
| | CBAM | 1.35 | 0.79 | 0.94 | 1.25 | 1.75 | 0.88 | 0.81 | 0.73 | 0.67 | 0.79 | 0.90 | 0.28 |
| | **SASE (ours)** | **1.38** | **0.84** | **1.07** | **1.38** | **1.89** | **1.20** | **1.41** | **0.92** | **1.21** | **1.21** | **1.18** | **0.52** |
| Coverage↑ | SLE | **1.0** | 0.96 | **1.0** | **1.0** | **1.0** | 0.95 | 0.95 | 0.93 | 0.95 | 0.98 | 0.89 | 0.85 |
| | SPAP | 0.86 | 0.90 | 0.92 | 0.94 | 0.95 | 0.88 | 0.84 | 0.79 | 0.71 | 0.68 | 0.92 | 0.81 |
| | CBAM | 1.0 | 0.95 | 1.0 | 1.0 | 1.0 | 0.91 | 0.88 | 0.83 | 0.78 | 0.73 | 0.90 | 0.83 |
| | **SASE (ours)** | **1.0** | **0.98** | **1.0** | **1.0** | **1.0** | **0.96** | **0.96** | **0.95** | **0.96** | **1.0** | **1.0** | **0.91** |

Table 1: Fidelity (FID, KID, Density) and diversity (Coverage) comparisons between our SASE and three baseline modules in low-shot image synthesis, including the vanilla SLE [6], the SPAP module [23] and the CBAM module [24], using the FastGAN pipeline [6] that utilizes the differentiable data augmentation (DiffAug) method [7] in training. Our SASE is consistently better than the three baseline modules.

**Model and Data Augmentation Baselines:** To evaluate the effectiveness of the proposed SASE, in addition to the vanilla SLE [6], we also compare with: the CBAM module [24] which leverages spatial and channel attention sequentially for better representation learning, and the SPAP module [23] which leverages multi-scale spatial attention in GANs.

For low-shot image synthesis, data augmentation plays an important role. The vanilla FastGAN [6] utilizes the differentiable data augmentation (DiffAug) method [7]. More recently, the adaptive data augmentation (ADA) method [25] is proposed with even better support for low-shot image synthesis. To evaluate whether the proposed SASE retains its effectiveness, we compare the SLE and our SASE in a modified FastGAN pipeline with the ADA in training. We follow the original ADA settings to set the target value to 0.6, and set the increasing rate of augmentation probability such that it can increase from 0 to 1 within 10k iterations (1/5 of the total training time).

| Metric | Method (**ADA**) | 256×256, ~100 images per category | | | | | 1024× 1024, ~1000 images | | | | 1024× 1024, ~100 images | | |
|---|---|---|---|---|---|---|---|---|---|---|---|---|---|
| | | Obama | Dog | Cat | Grumpy Cat | Panda | FFHQ | Art | Flower | Pokemon | AnimeFace | Skulls | Shells |
| FID↓ | SLE | 38.9 | 52.04 | 34.5 | 26.83 | 9.87 | 44.43 | 45.1 | 31.89 | 55.67 | 59.11 | 120.62 | 153.47 |
| | **SASE (ours)** | **34.5** | **49.83** | **31.2** | **26.03** | **9.50** | **39.12** | **43.53** | **29.63** | **48.56** | **53.31** | **96.56** | **140.75** |
| KID↓ | SLE | 0.01 | 0.015 | 0.004 | 0.007 | 0.003 | 0.012 | 0.011 | 0.006 | 0.013 | 0.018 | 0.049 | 0.071 |
| | **SASE (ours)** | **0.004** | **0.012** | **0.002** | **0.004** | **0.002** | **0.011** | **0.009** | 0.006 | **0.009** | **0.013** | **0.025** | **0.044** |
| Density↑ | SLE | 1.35 | 0.78 | 0.96 | 1.28 | 1.82 | 1.17 | 1.39 | 0.85 | 1.13 | 1.18 | 0.91 | 0.28 |
| | **SASE (ours)** | **1.39** | **0.89** | **1.12** | **1.41** | **1.87** | **1.21** | **1.40** | **0.92** | **1.25** | **1.24** | **1.21** | **0.53** |
| Coverage↑ | SLE | 1.0 | 0.94 | 1.0 | 1.0 | 1.0 | 0.96 | 0.94 | 0.95 | 0.94 | 0.98 | 0.92 | 0.86 |
| | **SASE (ours)** | 1.0 | **1.0** | 1.0 | 1.0 | 1.0 | 0.96 | **0.97** | **0.96** | **0.95** | **1.0** | **1.0** | **0.92** |

Table 2: Fidelity (FID, KID, Density) and diversity (Coverage) comparisons between our SASE and SLE using a modified FastGAN pipeline in which the differentiable data augmentation method is replaced by a more recent adaptive data augmentation method (ADA) [25] that facilitates low-shot image synthesis. Compared with Table 1, the ADA method shows better overall performance than the differentiable data augmentation method [7]. Our SASE remains consistently better than the SLE w.r.t. the new data augmentation method, justifying the architectural contributions by our SASE.

| Metric | DataAug | Method | 256×256, ~100 images per category | | | | | 1024× 1024, ~1000 images | | | | 1024× 1024, ~100 images | | |
|---|---|---|---|---|---|---|---|---|---|---|---|---|---|---|
| | | | Obama | Dog | Cat | Grumpy Cat | Panda | FFHQ | Art | Flower | Pokemon | AnimeFace | Skulls | Shells |
| KS $p$-value↑ (threshold 0.01) | DiffAug | SLE | 0.87 | 0.77 | 0.32 | 0.19 | 0.81 | 0.39 | 0.025 | 0.06 | 0.78 | 0.75 | 0.17 | 0.89 |
| | | SPAP | 0.45 | 0.39 | 0.13 | 0.11 | 0.55 | 0.12 | 0.11 | 0.03 | 0.42 | 0.31 | 0.21 | 0.49 |
| | | CBAM | 0.86 | 0.65 | 0.35 | 0.53 | 0.79 | 0.24 | 0.23 | 0.02 | 0.37 | 0.42 | 0.58 | 0.46 |
| | | SASE (ours) | 0.65 | 0.45 | 0.32 | 0.94 | 0.76 | 0.73 | 0.53 | 0.43 | 0.81 | 0.86 | 0.39 | 0.98 |
| | ADA | SLE | 0.83 | 0.74 | 0.35 | 0.19 | 0.84 | 0.37 | 0.027 | 0.08 | 0.76 | 0.73 | 0.18 | 0.90 |
| | | SASE (ours) | 0.71 | 0.59 | 0.42 | 0.95 | 0.81 | 0.74 | 0.55 | 0.42 | 0.83 | 0.87 | 0.38 | 0.98 |

Table 3: Memorization/overfitting assessment for our SASE and SLE, SPAP adn CBAM in the FastGAN pipeline with the differentiable data augmentation method. Our SASE shows no signs of overfitting across all scenarios, while the SLE shows tendency towards overfitting on some categories such as "Art" and "Flower".

**Results - Image Synthesis Quality:** Table 1 shows that the proposed SASE is consistently better than all baselines (SLE, SPAP and CBAM) in terms of both traditional metrics, FID and KID and more recently proposed more reliable metrics, Density and Coverage. For high-resolution (1024×A 1024) image synthesis which uses more SASE components (Fig. 3), the improvements are significantly

better, which shows the effectiveness of our SASE. It is also noted that the SLE is overall better than SPAP and CBAM.

Table 2 shows that our proposed SASE is still consistently better than the SLE when we replace the DiffAug by the ADA in training.

**Results - Memorization Assessment:** Table 3 shows that our SASE is capable of synthesizing new images by learning from low-shot images, while other methods have certain tendency towards overfitting on different categories. The results of our SASE are aligned with its diversity evaluation results in Table 1 and Table 2.

| Model | Nature-2k | Nature-5k | Nature-10k | FFHQ-2k | FFHQ-5k | FFHQ-10k |
|---|---|---|---|---|---|---|
| SLE | 103.71 | 104.73 | 99.64 | 27.68 | 20.6 | 19.21 |
| **SASE (ours)** | **101.53** | **96.91** | **93.94** | **24.59** | **19.45** | **18.84** |

Table 4: FID comparisons (smaller is better) on the 2 categories with models trained with more data (2k, 5k and 10k) at the resolution of $1024 \times 1024$.

| Model | Params | Resolution | FLOPs | Params | Resolution | FLOPs |
|---|---|---|---|---|---|---|
| SLE-FastGAN | 29.1M | 256×256 | **9.7933G** | 29.2M | 1024×1024 | **16.1960G** |
| SASE-FastGAN | 28.5M | 256×256 | 9.7942G | 28.5M | 1024×1024 | 16.1986G |

Table 5: Model complexity comparisons between SASE-FastGAN and SLE-FastGAN.

**Results - Learning with More Data:** To further evaluate the effectiveness of the proposed SASE when trained with more data, we compare our SASE and the SLE under different settings. With more training data, we use FID in evaluation for simplicity. As shown in Table 4, our SASE retains its stronger effectiveness than the SLE. Both models are trained from scratch with the same data sampled from the original FFHQ and Nature Photograph datasets. Training time are budgeted with around 20 hours for all the experiments.

**Qualitative Results and Explainability Visualization.** To check what are learned by the spatial attention (the Key maps in Fig. 3), Fig. 5 shows some synthesized face images and the learned latent spatial masks. We can see that the learned masks cover different areas of the face, e.g. starting from left, the fourth column of masks cover the hair area, and the third column covers nose area. **Please check the Appendix A.3 for more qualitative results.**

**Model Complexity.** Table 5 shows the comparisons of model sizes. Our SASE-FastGANs have slightly less parameters than the vanilla SLE-FastGANs, and has negligible computing cost increase interms of FLOPs. since we split and squeeze the channel dimension ($g$ and $r$ in right of Fig. 2) in learning the channel-wise and spatial attention

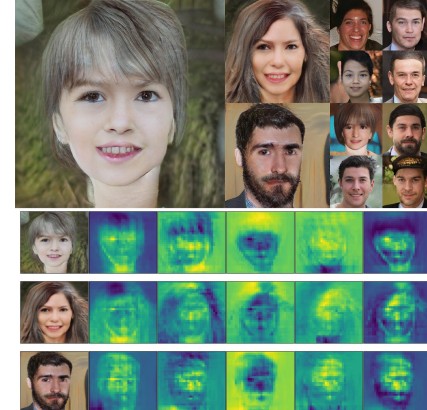

Figure 5: *Top:* Synthesized face images at the resolution of $1024 \times 1024$ in the FFHQ dataset [3]. The model is trained using 2k training FFHQ images for around 15 hours on a single GPU. *Bottom:* Visualization of the learned Key maps (spatial masks) from the stage $32^2$ to the stage $512^2$.

in our SASE. Although having less number of parameters, the proposed SASE module increases the training time in training models for high-resolution image synthesis (roughly $1/8$ relative increase), which may due to the more sophisticated computational graphs to maintain for forward and backward computation after the SASE is used. The training time increase is negligible for training image synthesis models at lower resolutions such as $256 \times 256$.

### 3.2 One-Shot Image Synthesis Results

**Data.** Since our goal is to test if the proposed SASE can lead to structure-aware one-shot image synthesis, we select 23 images, among which 14 images are used in the vanilla SinGANs: Brandenberg, bridge, Golden gate, tower, angkorwat, balloons, birds, colusseum, mountains, starry night, tree, cows, volcano; The remaining 9 images are searched from the website. The images cover different structures which often fail the vanilla SinGANs.

**Settings and Baselines.** We follow the settings provided by the vanilla SinGANs [12]. Two baselines are used: (i) *ConSinGANs* [26] for

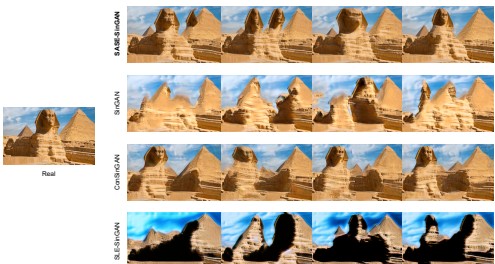

Figure 6: *Left:* a real image. *Right:* from top to bottom, synthesized images by our SASE-SinGAN, the vanilla SinGAN, the ConSinGAN, and the SLE-SinGAN. SASE-SinGAN is better in terms of preserving structure, while producing meaningful semantic variations (e.g., the change of number of Sphinx statues)

273 which we follow the suggestions in the paper to try different combinations between the learning rate
274 and the number of stages jointly trained and select the best results. (ii) *SLE-SinGANs* in which the
275 SLE module is used, instead of SASE, in the right-bottom of Fig. 3.

276 **Metrics.** We evaluate our methods with single image FID (SIFID) and Diversity Score proposed in
277 the vanilla SinGANs [12].

| Metric | SinGAN | ConSinGAN | SLE-SinGAN | SASE-SinGAN (**ours**) |
|---|---|---|---|---|
| SIFID$\downarrow$ | 0.683 | 1.45 | 0.78 | **0.581** |
| Diversity$\uparrow$ | 0.543 | 0.487 | **0.559** | 0.295 |

Table 6: SIFID and diversity score comparisons using the 23 selected images. Note that SIFID may not reflect the actual quality of synthesized images, as pointed out in ConSinGANs [26].

| Model | Params | Resolution | FLOPs | Params | Resolution | FLOPs |
|---|---|---|---|---|---|---|
| SinGAN | 1.02M | 165×250 | 19.33G | 1.02M | 330×250 | 38.20G |
| SLE-SinGAN | 1.63M | 165×250 | 19.33G | 1.63M | 330×250 | 38.21G |
| SASE-SinGAN | 1.25M | 165×250 | 22.85G | 1.25M | 330×250 | 44.65G |
| ConSinGAN | **0.77M** | 165×250 | **8.92G** | **0.77M** | 330×250 | **17.73G** |

Table 7: Model complexity comparison between the proposed SASE-SinGAN and other SinGAN variants.

278 **Results.** Table 6 shows the comparison results. In terms of diversity score, Our SASE obtains lower
279 diversity in the trend similar to ConSinGAN. The testing images are structure-rich images for which
280 our goal is to study how to preserve the structure. The diversity score should be interpreted jointly
281 with the SIFID. Fig. 6 shows synthesized images for the Egyptian pyramid image. We can see the the
282 proposed SASE is stronger in terms of preserving structures in synthesized images. We observe that
283 ConSinGANs may fail to learn some images e.g., the Golden Gate (Fig. 23 in the appendix), which
284 causes the high SIFID. **More qualitative results are in the Appendix A.4**.

285 **Model Complexity**. Table 7 shows the complexity comparison between SASE-SinGAN and other
286 SinGAN variants. Both SLE and our SASE increases the FLOPs significantly compare to vanilla
287 Singan, since they are used for connecting low-resolution feature maps to relatively high ones and
288 there are four in total, see Fig. 3, so the overhead is light-weight. For SinGANs, we have SLE and
289 SASE between feature maps with the same resolution and have one at every resolution stage. The
290 spatial attention branch of our SASE increases the FLOPs even more significantly.

### 3.3 Image Classification Results in ImageNet-1000

292 We test the proposed SASE using ResNet-
293 50 [8] (Fig. 4). First we compare it with SE
294 and other variants of attention models used in
295 ResNets, including the Self-Calibrated convo-
296 lutions (SC-ResNets) [27], the Gather-Excite
297 networks (GE-ResNets) [28], the GC-ResNets
298 (non-local networks meet the SE networks) [29],
299 the Efficient Channel Attention networks (ECA-
300 ResNets) [30], and the attention augmented net-
301 works (AA-ResNets) [31]. For fair comparisons,
302 we use the most vanilla training settings to verify
303 the effectiveness of the architectural design of
304 SASE itself: 100 epochs and the basic data aug-
305 mentation scheme (random crop and horizontal
306 flip). Table 8 (top) shows the results. Compared

| Epochs | Method | #Params$\downarrow$ | FLOPS$\downarrow$ | top-1$\uparrow$ | top-5$\uparrow$ |
|---|---|---|---|---|---|
| 100 | [†]SE-ResNet50 | 28.09M | 4.13G | 77.74 | 93.84 |
| | [†]ResNeXt-32x4d-50 | 25.03M | 4.27G | 77.90 | 93.66 |
| | SC-ResNet50 [27] | 25.60M | 4.00G | 77.80 | 93.90 |
| | GE-ResNet50 [28] | 31.20M | 3.87G | 78.00 | 94.13 |
| | GC-ResNet50 [29] | 28.08M | 3.87G | 77.70 | 93.66 |
| | ECA-ResNet50 [30] | 24.37M | 3.86G | 77.48 | 93.68 |
| | AA-ResNet50 [31] | 25.80M | 8.30G | 77.70 | 93.80 |
| | SASE-ResNet50 (**ours**) | **18.66M** | **3.36G** | 78.06 | 94.14 |
| 300 | Swin-Tiny [32] | 28.00M | 4.50G | 81.20 | **95.50** |
| | PVT-Small [33] | 24.50M | 3.80G | 79.80 | - |
| | ResNet50-Strikesback (A2) [34] | 25.60M | 4.10G | 79.80 | - |
| | SASE-ResNet50 (A2) (**ours**) | 18.66M | 3.36G | **81.24** | 95.34 |
| 200 | BoT-S1-50 [17] | 20.80M | 4.27G | 79.10 | 94.40 |

Table 8: Comparisons of ImageNet-1000 classification results. All models are trained and tested using the resolution of $224 \times 224$. Top-1 and Top-5 accuracy (%) are used. [†] Results are from the MMClassification model zoo.

307 with the SE, our SASE obtains 0.32% top-1 accuracy increase, while significantly reducing the model
308 parameters (by 9M) and FLOPs. Compared with ResNeXt-32x4d-50, our SASE obtains 0.16% top-1
309 accuracy increase with much less parameters too, and our SASE also outperforms other attention
310 variants. These results clearly show the effectiveness of the proposed SASE.

311 Further, to compare the recent state of the art image classification models, we follow the improved
312 training procedure, the A2 recipe, proposed in [34] that enables ResNets to strike back in performance
313 compared with variants of Vision Transformer, the results are shown in Table 8 (middle). The
314 proposed SASE shows very promising performance, bridging the performance gap between the
315 ResNets and the state-of-the-art Swin-Transformer [32] and Pyramid Vision Transformer (PVT) [33],
316 which supports our design hypothesis stated in Section 2.2.

### 3.4 Object Detection and Instance Segmentation in MS-COCO

318 To check how well the ImageNet-100 pretrained SASE-ResNet50 will transfer to downstream
319 tasks, we test it in the MS-COCO 2017 object detection and instance segmentation dataset [35]
320 using the Mask R-CNN framework [18]. Table 9 shows the comparisons. On the one hand, our

SASE significantly outperforms the vanilla ResNet, and is slightly better than the Bottleneck Transformer [17] with a smaller model complexity, which shows the effectiveness of our SASE. On the other hand, our SASE obtains comparable performance to the PVT-Small [33], but is significantly worse than the Swin-T [32].

| Backbone | Mask R-CNN 3× (36 epochs) | | | | | | |
|---|---|---|---|---|---|---|---|
| | #P(M) | $AP^b$ | $AP^b_{50}$ | $AP^b_{75}$ | $AP^m$ | $AP^m_{50}$ | $AP^m_{75}$ |
| ResNet50 | 44.2 | 41.0 | 61.7 | 44.9 | 37.1 | 58.4 | 40.1 |
| PVT-Small [33] | 44.1 | 43.3 | 65.3 | 46.9 | 39.9 | 62.5 | 42.8 |
| BoT50 [17] | - | 43.6 | 65.3 | 47.6 | 38.9 | 62.5 | 41.3 |
| Swin-T [32] | 48.0 | **46.0** | 68.1 | 50.3 | **41.6** | 65.1 | 44.9 |
| SASE-ResNet50 (ours) | **37.5** | 43.7 | 65.2 | 47.7 | 39.5 | 62.0 | 42.3 |

Table 9: Performance comparisons in MS-COCO.

## 4 Related Work

**Light-weight GANs with Low-Shot Learning.** Compared to the extensive research on light-weight neural architectures in discriminative learning for mobile platforms, much less work has been done in generative learning [36]. The residual network [8] is the most popular choice, on top of which powerful generative models such as BigGANs [2] and StyleGANs [37, 4] have been built with remarkable progress achieved. For high-resolution image synthesis, these models will be very computational expensive in training and inference. In the meanwhile, training these models typically require a big dataset, which further increase the training time. low-shot learning is appealing, but very challenging in training GANs, since data augmentation methods that are developed for discriminative learning tasks are not directly applicable. To address this challenge, differentiable data augmentation methods and variants [7, 25, 38, 39] have been proposed in training GANs with very exciting results obtained. Very recently, a FastGAN approach [6] is proposed to realize light-weight yet sufficiently powerful GANs with several novel designs including the SLE module. The proposed SASE is built on the SLE in FastGANs to reinforce its data-specificity.

**Learning Unconditional GANs from a Single Image.** There are several work on learning GANs from a single texture image [40, 41, 42]. Recently, a SinGAN approach [12] has shown surprisingly good results on learning unconditional GANs from a single non-texture image. It is further improved in ConSinGANs [26] which jointly train several stages in progressively growing the generator network. However, it remains a challenging problem of preserving image structure in synthesis. The proposed SASE is applied to the vanilla SinGANs [12], leading to a simplified workflow that can be trained in a stage-wise manner and thus more efficient than ConSinGANs, and facilitating a stronger capability of preserving image structures.

**Attention Mechanism in Deep Networks.** Attention reallocates the available computational resources to the most relevant components of a signal to the task [43, 44, 45, 46, 47, 11]. Attention mechanisms have been widely used in computer vision tasks [48, 49, 50, 51]. The SE module [9] applies a lightweight self gating module to facilitate channel-wise feature attention. Our proposed SASE module incorporates spatially-adaptive feature modulation, while maintaining the light weight design, improving the representation power for efficient discriminative learning.

## 5 Limitations and Potential Negative Impacts of the Proposed Work

The proposed SASE shows worse performance in object detection and instance segmentation than state-of-the-art Transformer based models. One direction to address this is to run more comprehensive experiments on the design choices (Eqn. 9 to Eqn. 12). The proposed SASE module does not show any potential negative impacts with its current form.

## 6 Conclusion

This paper proposes to learn spatially-adaptive squeeze-excitation (SASE) networks for better data-specificity by jointly learning both channel-wise attention as latent style representation and spatial attention as latent layout representation. The resulting SASE module computes a 3D attention matrix for modulating an input feature map. In experiments, the proposed SASE module is tested in low-shot image synthesis using FastGANs, one-shot image synthesis using SinGANs, ImageNet-1000 classification using ResNets, MS-COCO object detection using Mask R-CNN. The SASE-FastGANs are consistently better than three strong baselines, and obtain significantly better performance at high-resolution image synthesis. The SASE-SinGANs show stronger capabilities in preserving image structures than prior arts. The SASE-ResNets show better performance than the SE variant and other variants with significantly smaller models, and competitive performance to state-of-the-art Transformer based models.

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

## A  Appendix

### A.1  Comparing the SASE with Alternative Designs in Image Synthesis

*Comparing with the weight modulation in StyleGANv2* [4]. The weight modulation method in StyleGANv2 is an elegantly designed operation to achieve detailed style tuning effects. The style code is used to directly modulate the filter kernels (as model parameters) in an instance specific, and then modulated filter kernels are used in computing the convolution. Although being highly expressive, this weight modulate is not spatially-adaptive. And, it increases the computational burden and the memory footprint in execution. The proposed SASE directly modulates the feature map in a light-weight manner.

*Comparing with the SPADE in GauGANs [52] and the ISLA-Norm in LostGANs [53].* Both the SPADE and the ISLA-Norm exploit spatially-adaptive modulation, but apply it inside the BatchNorm. They replace the vanilla channel-wise affine transformation in the BatchNorm with spatially-adaptive affine transformation. The spatially-adaptive affine transformation coefficients are learned either from the input semantic masks in GauGANs or the generated latent masks from the input layouts in LostGANs. The proposed SASE is similar in spirit to the ISLA-Norm, but is formulated under the Inception architecture together with the skip-layer idea proposed in FastGANs [6].

### A.2  Training details of SASE-FastGANs

We use the Adam optimizer for training, with $\beta_1$=0.5, $\beta_2$=0.999. We use learning rate of 0.0002 for all datasets except for FFHQ and the panda datasets, in which we use 0.0001. For the architecture of Discriminator, we adopts the output size of $5 \times 5$; and for SASE-FastGAN model on the $1024 \times 1024$ datasets, we apply Gaussian noise injection to the spatial masks of SASE (the right-bottom of Fig. **??**), with zero mean and unit variance; For the convolution of spatial branch of SASE, we set the dilation rates as 2, 2, 4 at stage $8 \times 8$, $16 \times 16$, $32 \times 32$, respectively.

### A.3  More results of SASE-FastGANs

#### A.3.1  Clarification on results on FFHQ-1k

We notice there is a gap of the performance on FFHQ-1K between our retrained version based on the official FastGAN code and the reported one in the paper. Thanks to the author's feedback via emails, the best configuration for reproducing the FFHQ-1k result of FID=24.45 will NOT be released since it is deployed on a commercial platform. So the FID of the SLE-FastGAN we trained based on the FastGAN code is worse than the one reported in the paper (Table 10).

| | FID |
|---|---|
| SLE-FastGAN [6] (reported in the paper) | 24.45 |
| Retrained from the official FastGAN code | 44.31 |
| Retrained from the official FastGAN code (dataset-specific version) | 42.82 |
| Our SASE-FastGAN built on the official FastGAN code | 39.59 |

Table 10: FFHQ-1k performance comparison between the reported result in FastGAN paper, our retrained version based on their code, and the proposed SASE-FastGAN

#### A.3.2  Synthesis results of SASE-FastGAN on $1024 \times 1024$ datasets

Fig. 7 shows the example synthesized $1024 \times 1024$ images of our proposed SASE-FastGAN.

#### A.3.3  Backtracking results

**Settings:** (Thanks to the clarification by the authors of FastGANs via emails) 1) We first split the dataset into train/test ratio of 9:1. 2) Train the model on the splitted training set. 3) Pick the trained generator checkpoint at iteration (20k, 40k, 80k) respectively, and do latent backtracking for 1k iterations on test set. 4) Compute the mean LPIPS between the test images and the reconstructed images from backtracking of the corresponding checkpoints. Where LPIPS is the average perceptual distance between two set of images; in this test, a lower LPIPS value indicates less overfitting, since

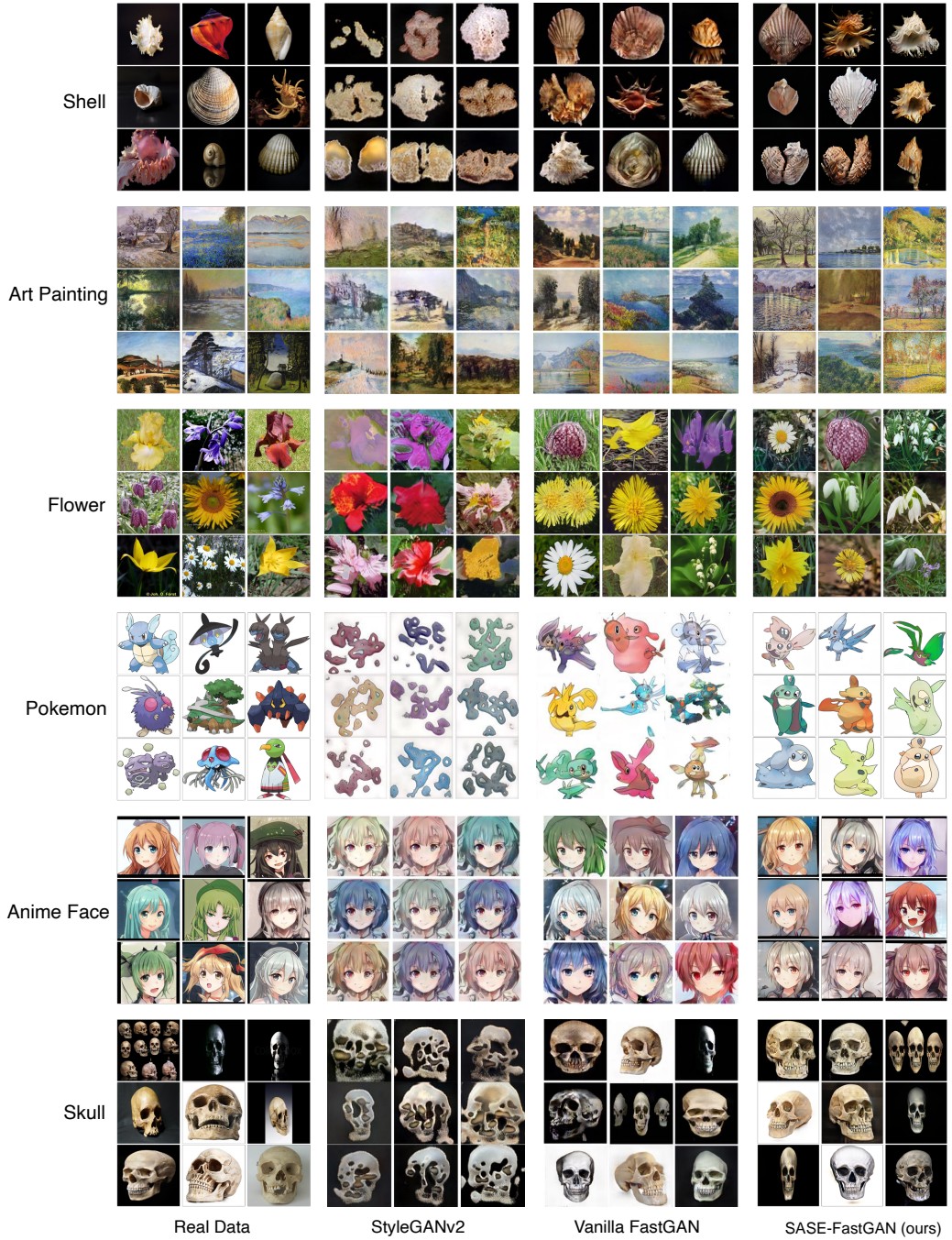

Figure 7: Examples of synthesized images at the resolution of $1024 \times 1024$. Best viewed in magnification.

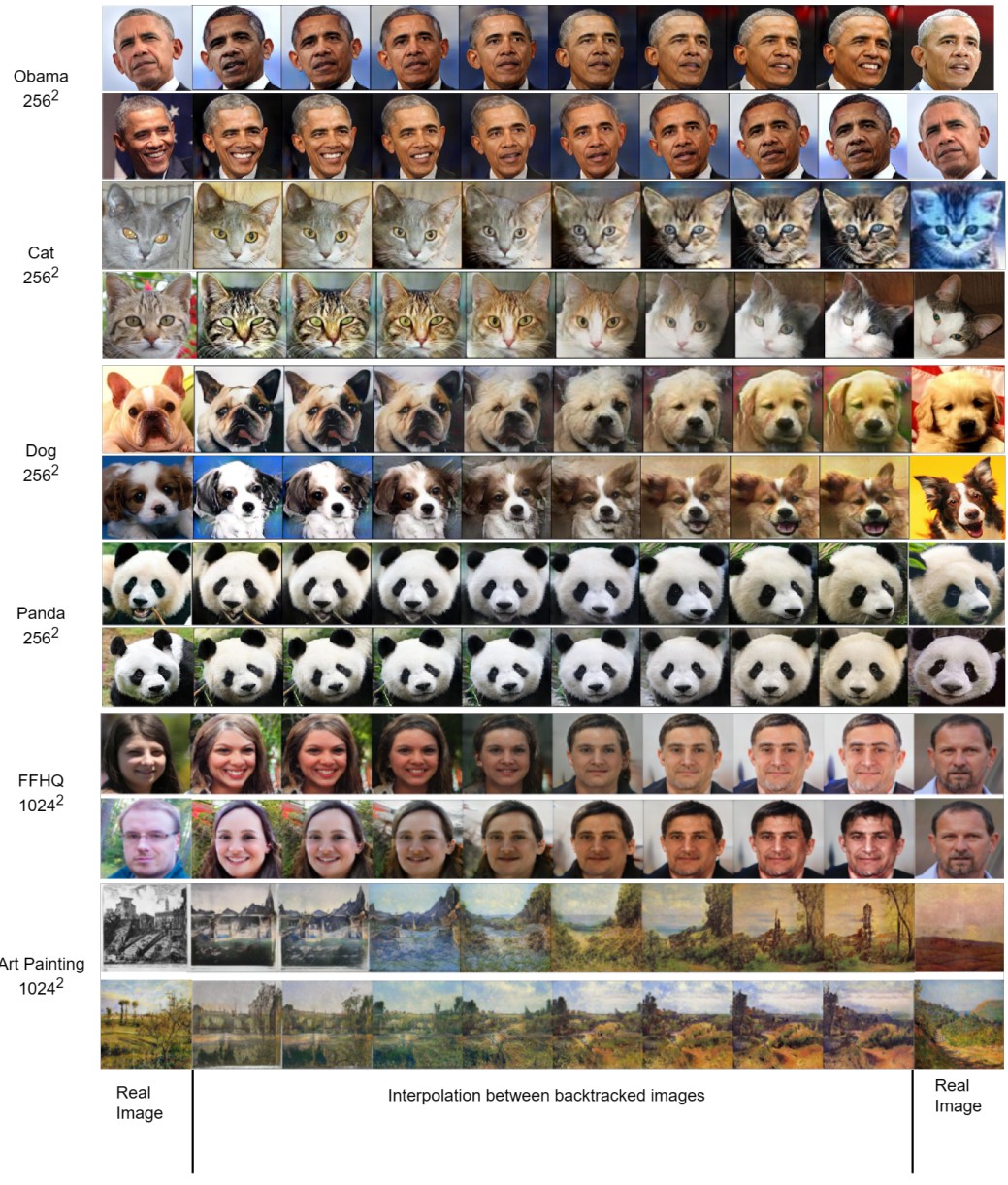

Figure 8: Examples of backtracking results,

it means that the model trained on the training set can backtrack images on an unseen testset with small reconstruction error.

**Results:** Fig. 8 shows the example backtracking results on several of the low-shot datasets. The smooth transition of the interpolated images between the backtracked test images show that our model hasn't overfit to the training set.

### A.3.4 Style mixing results

**Settings:** To demonstrate that the proposed SASE is able to disentangle the high level semantic attributes of featres at different scales, we conduct the style mixing experiment as done in the FastGAN paper [6], in which for a pair of style and content images, we extract channel weights from style images, and use them to modulate the features of content images, while retaining the spatial masks of the content images. The resulting effects as shown in Fig. 9 is that the appearance and color scheme of the style image is propagated to the content image, and the spatial structure of the content image is unchanged.

### A.4 More results of SASE-SinGANs

In order to demonstrate the strength of the proposed SASE module in one-shot generative learning. We present qualitative comparison of synthesis results of SASE-SinGAN with other related methods on 9 images, which are buildings that have different global structures. We also show the results of image harmonization and editing under one-shot setting.

### A.4.1 Synthesis with example images

Fig. 10 to Fig. 23 are the example synthesis results. We can see that compare to ConSinGAN, SinGAN and SLE-SinGAN, SASE-SinGAN captures the global layout of the image better, while producing meaningful local semantic variations, also notice that ConSinGAN fails to learn some of the image, as shown in Fig. 23.

### A.4.2 Harmonization

Fig. 24 shows the comparison on one-shot image harmonization task as done in [12]. It shows that our proposed SASE-SinGAN can realistically blend an object into the background image.

### A.4.3 Editing

Fig. 25 shows the comparison on one-shot image editing task as done in [12]. It shows that our proposed SASE-SinGAN is able to produce a seamless composite in which image regions have been copied and pasted in other locations. Note that SASE-SinGAN shows more realistic composite within the edited regions.

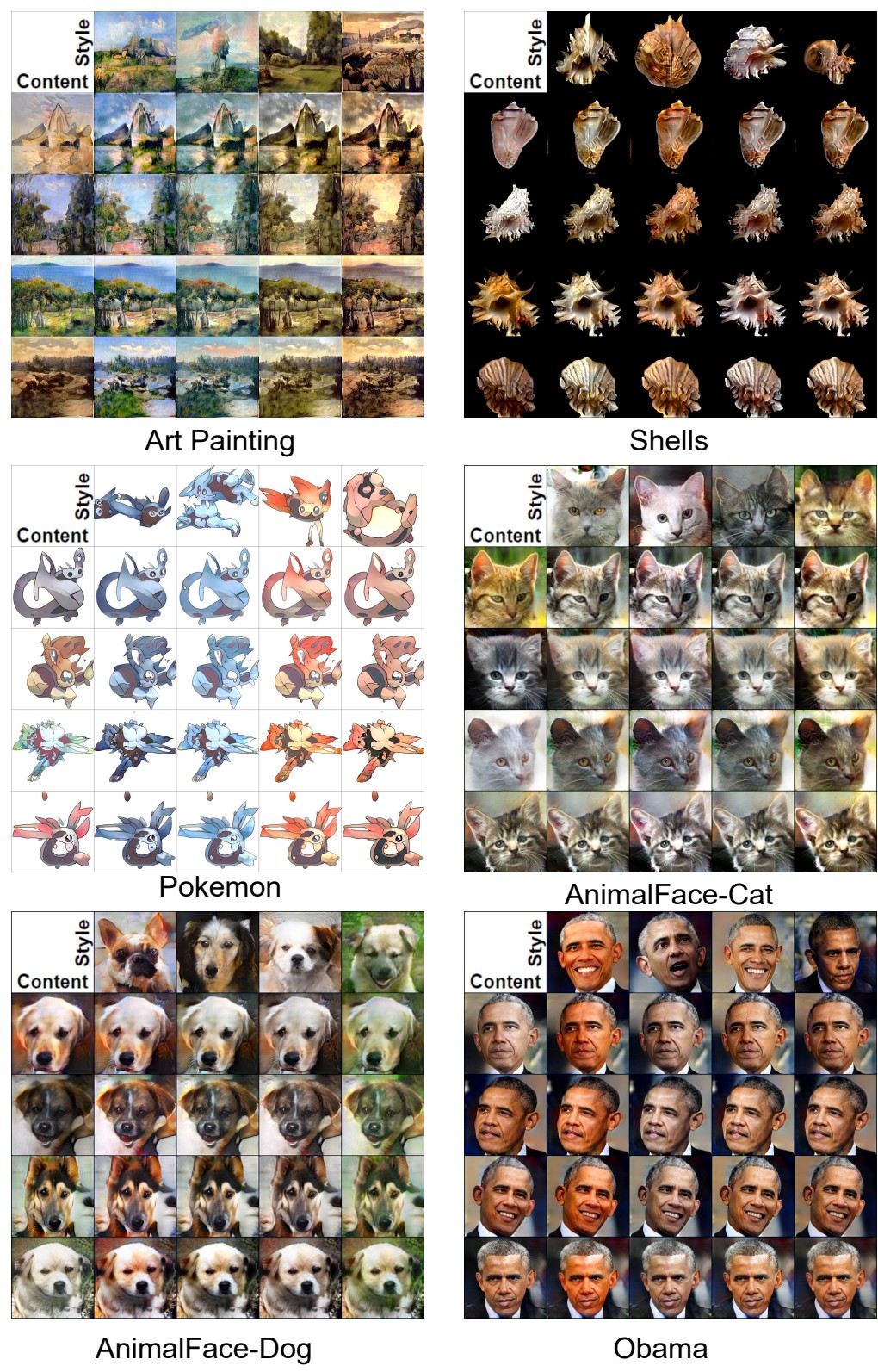

Figure 9: Examples of style mixing results. Best viewed in magnification.

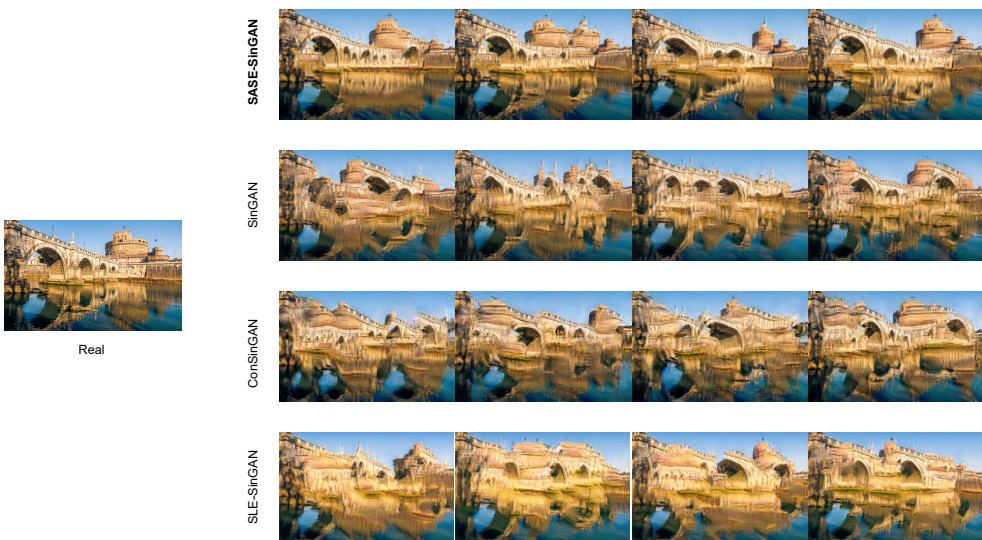

Figure 10: One-Shot synthesis comparison on the bridge image. Note how the synthesized images of SASE-SinGAN capture the global layout of the real image, and at the same time produces semantically meaningful variations (size, number of towers at top).

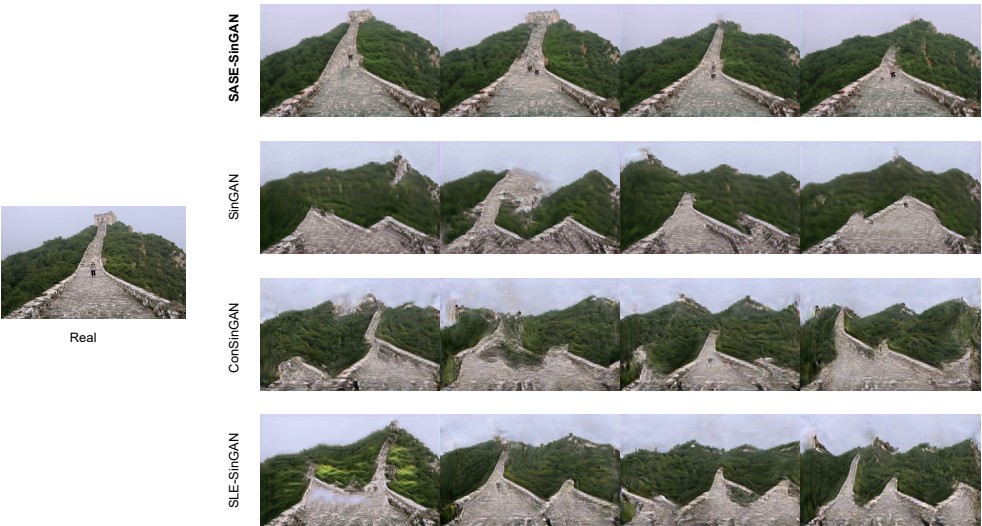

Figure 11: One-Shot synthesis comparison on the Great Wall image.

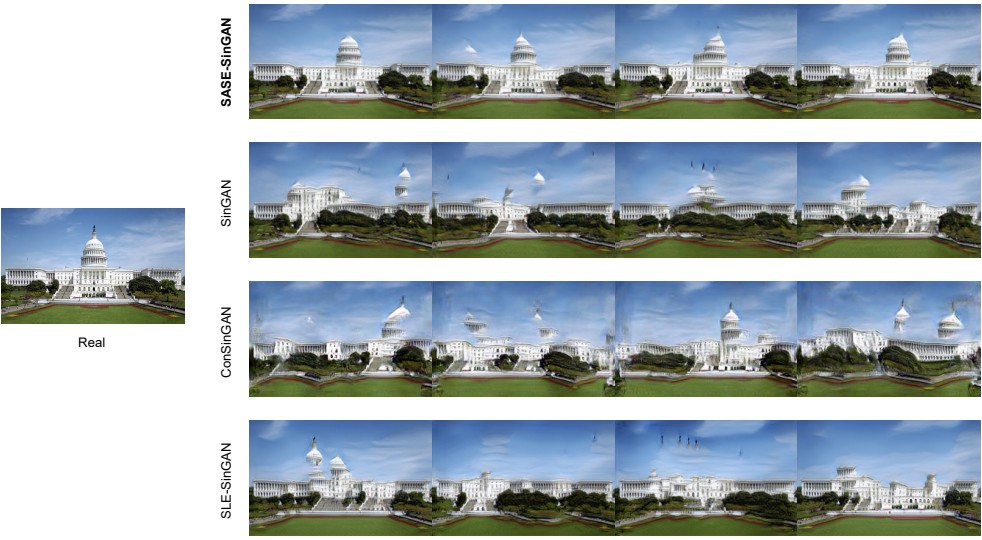

Figure 12: One-Shot synthesis comparison on the capitol hill image.

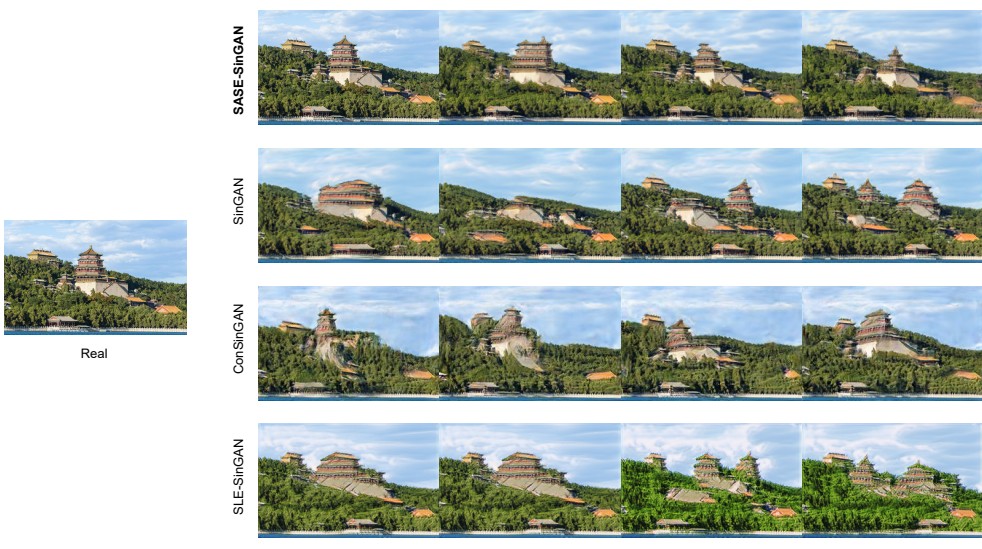

Figure 13: One-Shot synthesis comparison on the ancient Chinese tower image.

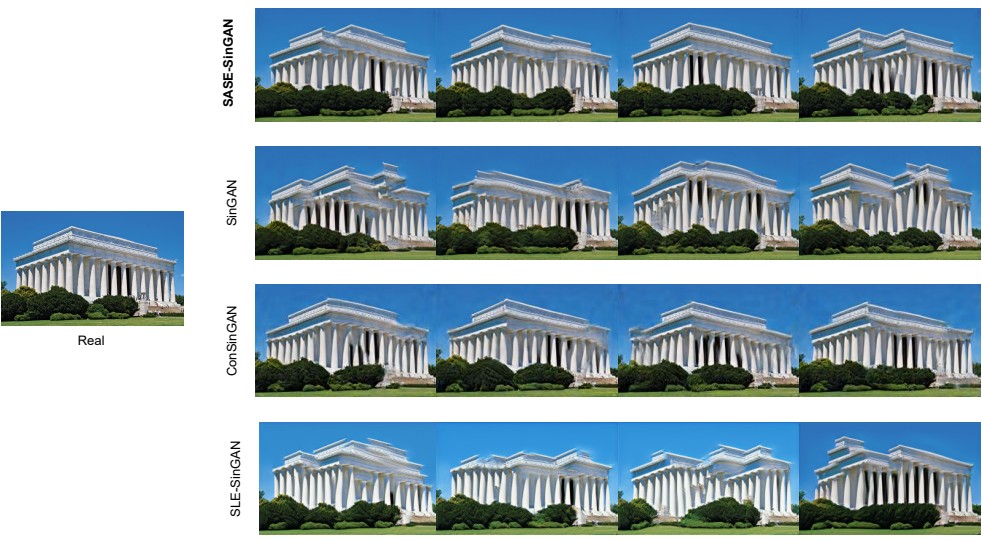

Figure 14: One-Shot synthesis comparison on the Lincoln memorial image.

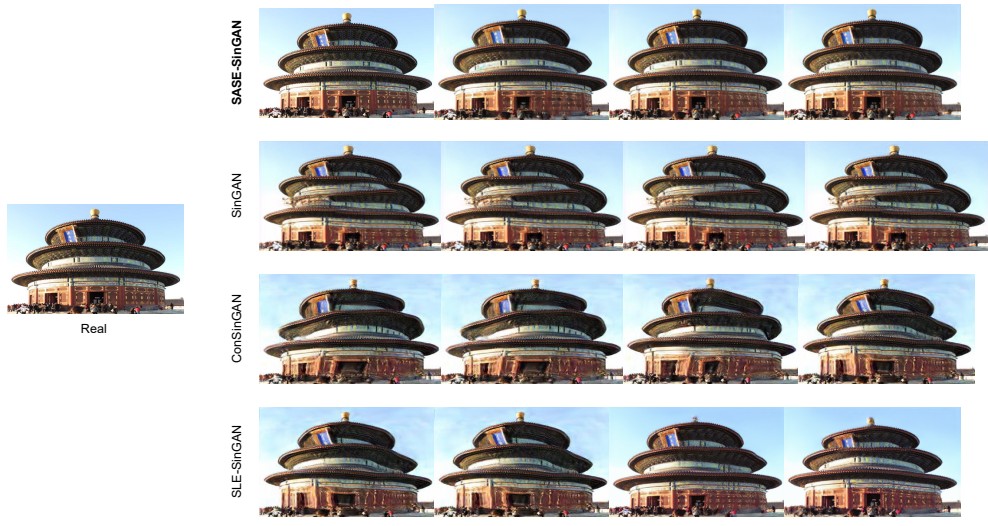

Figure 15: One-Shot synthesis comparison on the Temple of Heaven image.

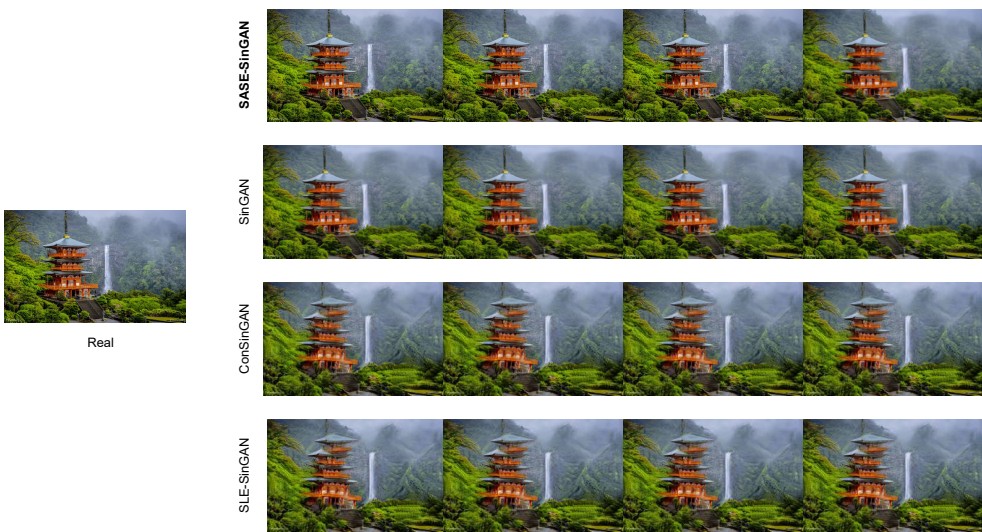

Figure 16: One-Shot synthesis comparison on the Temple of ancient Chinese tower image.

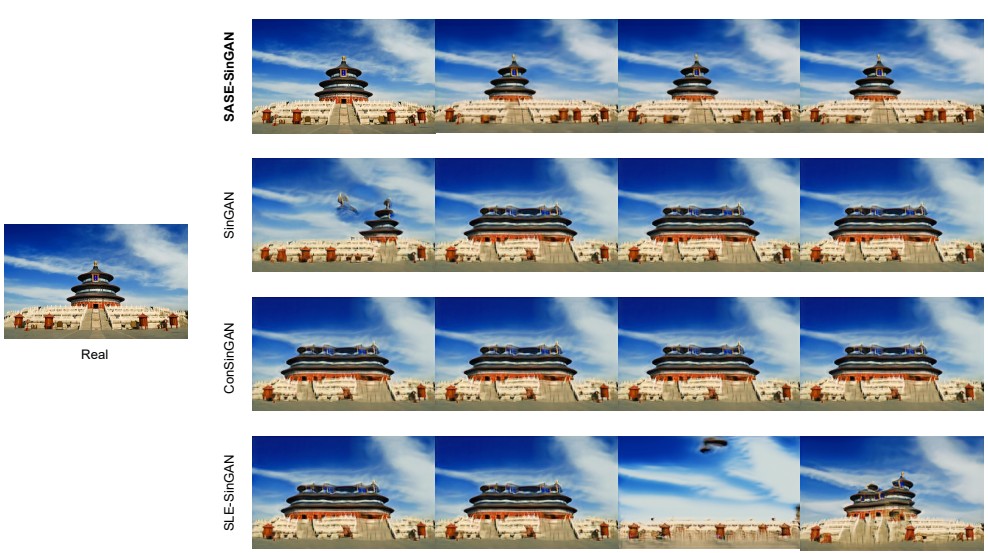

Figure 17: One-Shot synthesis comparison on the Temple of Heaven image.

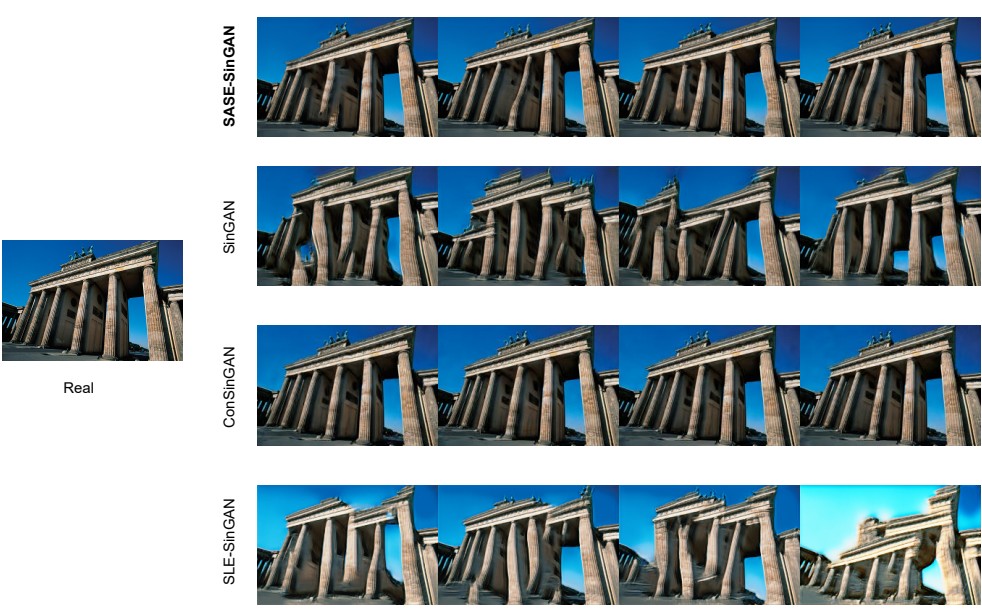

Figure 18: One-Shot synthesis comparison on the Brandenberg image.

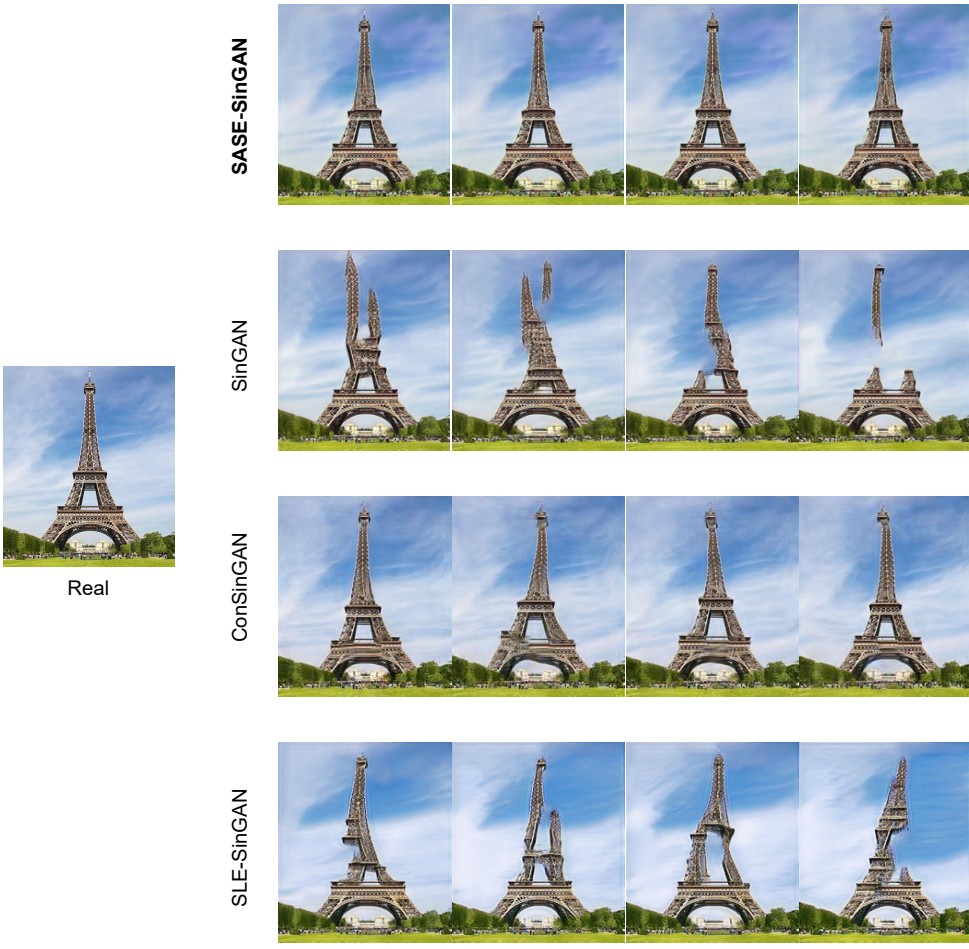

Figure 19: One-Shot synthesis comparison on the Eiffel tower image.

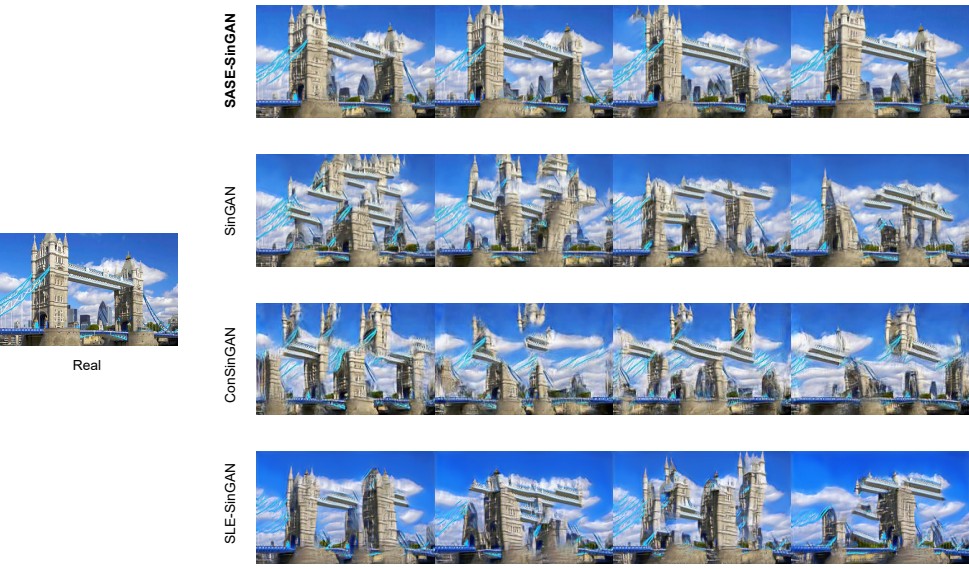

Figure 20: One-Shot synthesis comparison on the bridge image.

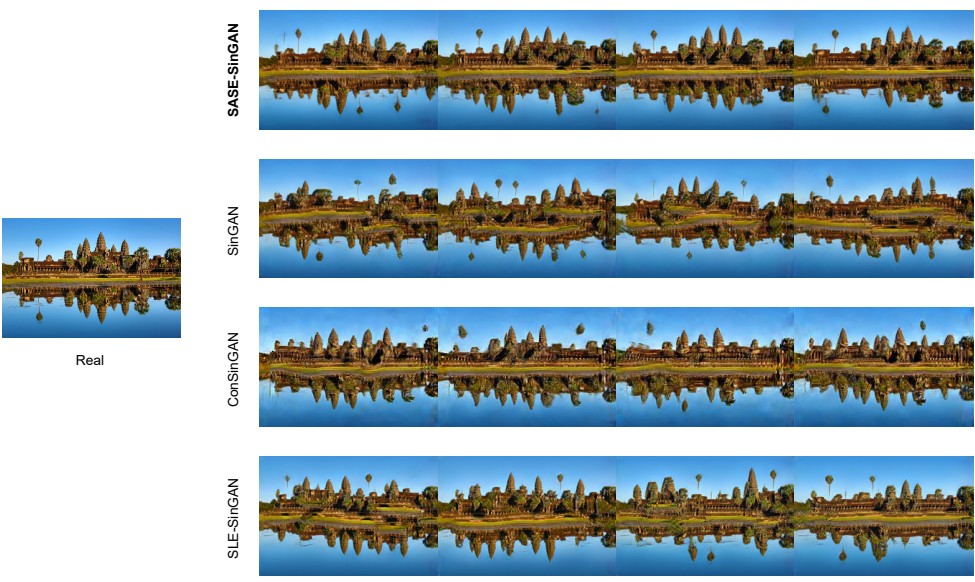

Figure 21: One-Shot synthesis comparison on the angkorwat image.

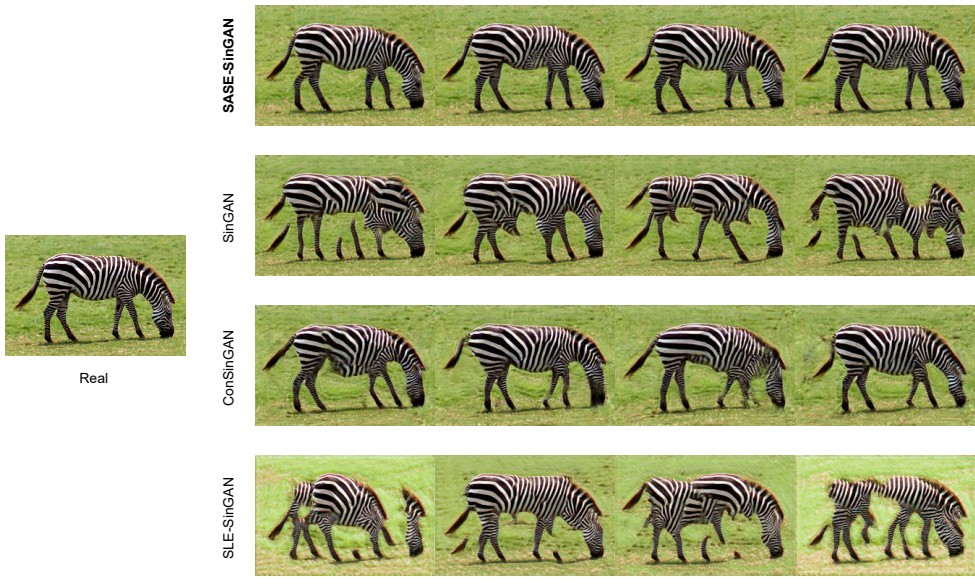

Figure 22: One-Shot synthesis comparison on the zebra image.

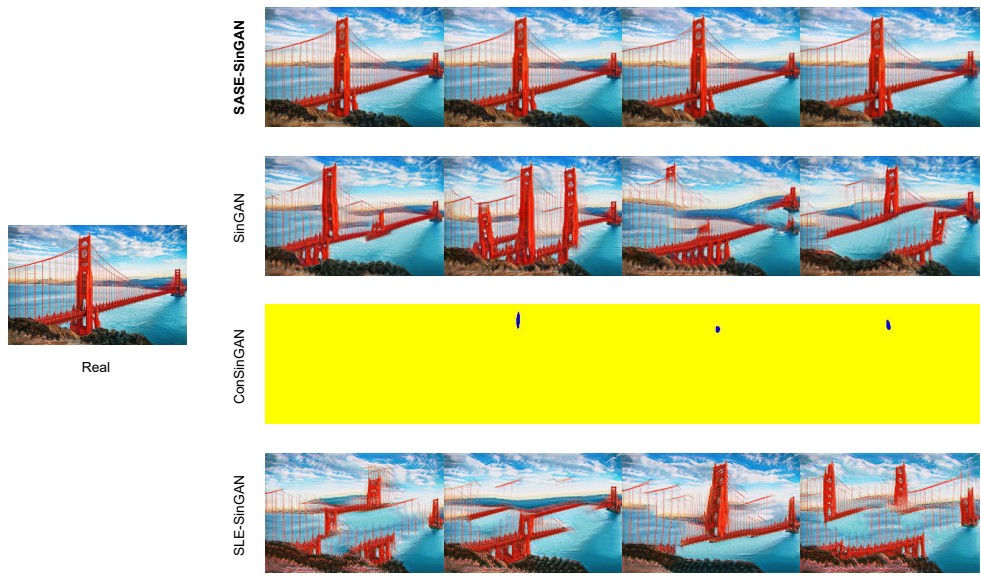

Figure 23: One-Shot synthesis comparison on the Golden Gate image. Notice that training of the ConSinGAN has collapsed.

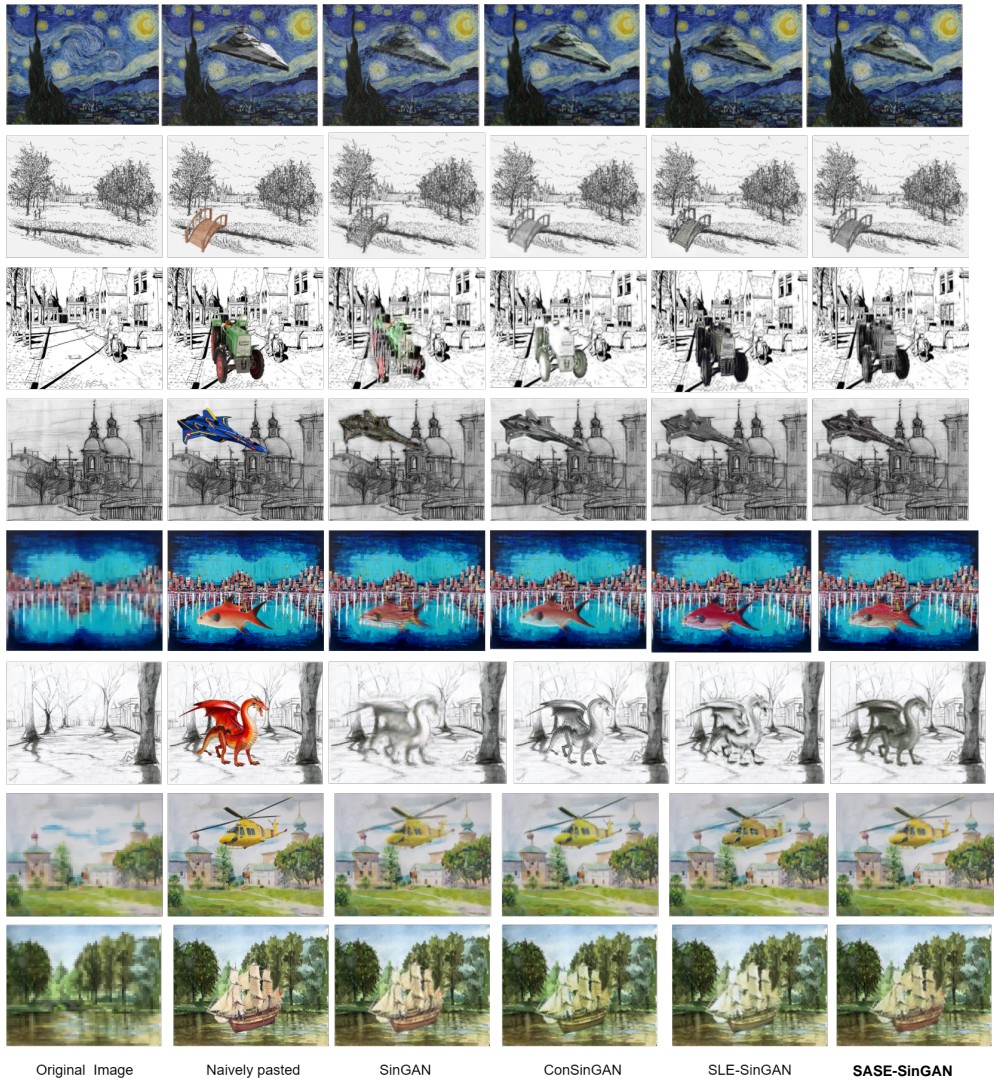

Original  Image          Naively pasted          SinGAN          ConSinGAN          SLE-SinGAN          **SASE-SinGAN**

Figure 24: One-Shot harmonization comparison on example images.

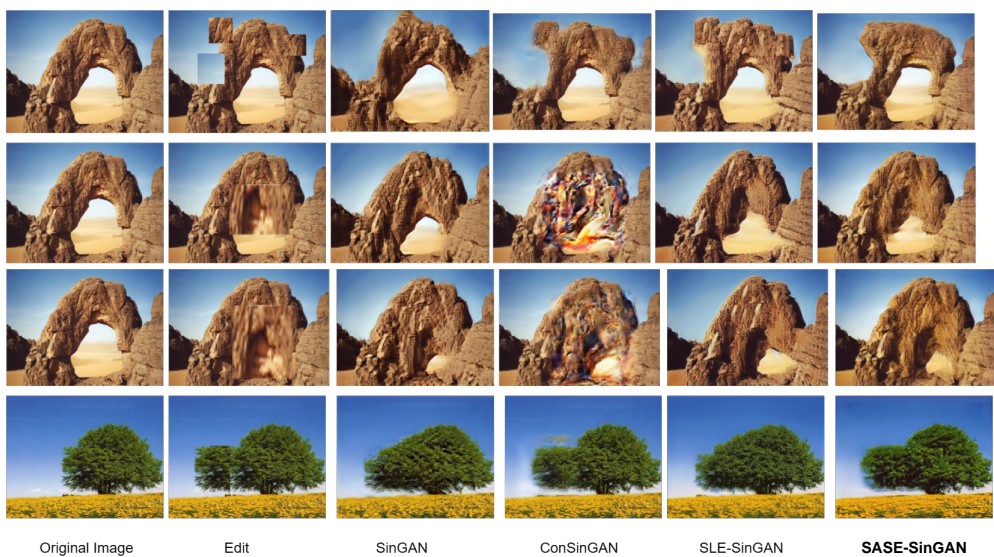

Original Image      Edit      SinGAN      ConSinGAN      SLE-SinGAN      **SASE-SinGAN**

Figure 25: One-Shot Editing comparison on example images.

