# OpenReview forum: "Learning Spatially-Adaptive Squeeze-Excitation Networks for Image Synthesis and Image Recognition"
_NeurIPS.cc/2022/Conference — NeurIPS 2022 Submitted_

### Official Review · Reviewer_bFtN · 2022-07-04

**Rating:** 5
**Confidence:** 4
**Soundness:** 3 good
**Presentation:** 3 good
**Contribution:** 3 good

**Summary:**

This paper proposes a Spatially-Adaptive Squeeze-Excitation (SASE) module which aims to learn a full attention mechanism as a 3D attention matrix. Two variants are developed to address the image classification and image synthesis tasks. Experimental results on public datasets show its performance.

**Questions:**

There are some issues which are listed as follows.

1, The idea of SASE is modified based on SE to include the spatial attention mechanism, thus it can learn the so-called full attention as the form of 3D matrix. While this idea is similar to the dual attention mechanism [a], which learns attention matrixes to consider spatial attention and channel attention at the same time.

[a] Dual Attention Network for Scene Segmentation.

2. The experimental results are not sufficiently strong. For example, in table 8, the improvement over the existing methods is only tiny. At the same time, in table 6, the proposed method achieves the worst diversity performance, is there any reason for that?

**Limitations:**

Please see the above sections.

**Strengths And Weaknesses:**

+ The paper is easy to follow. Its representation is clear.
+ The idea of learning a full attention matrix is interesting.
+ Extensive experimental studies are conducted.

- The experimental results are not strong.
- The idea is essentially similar to the dual attention mechanism.

---

> ### Author Response · Authors · 2022-08-02
> **Response 1/1**
>
> Thank you for your time and efforts reviewing our paper.
>
> **Comments**:  The idea of SASE is modified based on SE to include the spatial attention mechanism, thus it can learn the so-called full attention as the form of 3D matrix. While this idea is similar to the dual attention mechanism [a], which learns attention matrixes to consider spatial attention and channel attention at the same time.
>
> **Response**:  Thank you for pointing out the dual attention paper. We will discuss it in the revision.  In the dual attention paper, the spatial attention and the channel attention are fused by summation. The spatial attention is more similar to the self-attention in Transformer using a quadratic position aware matrix ((HxW) x (HxW). In our SASE,  the spatial and channel attention are coupled via product (instead of summation), and we use much more light-weight design (Fig.2, Eqn.6 to 13).
>
>
> **Comments**: The experimental results are not sufficiently strong. For example, in table 8, the improvement over the existing methods is only tiny. At the same time, in table 6, the proposed method achieves the worst diversity performance, is there any reason for that?
>
> **Response**:  In Table 8, for image classification in ImageNet-1k,  with 100 epochs in training, the improvement made by our SASE is nontrivial, especially given that our SASE uses much less parameters, as well as less FLOPs.  With 300 epochs, our SASE-ResNet50 significantly outperforms ResNet50-Strikeback, and can even slightly better than two state-of-the-art Transformer models (Swin-Tiny and PVT-Small).
>
> In Table 6, as explained in lines 279-281, "in terms of diversity score, Our SASE obtains lower diversity in the trend similar to ConSinGAN. The testing images are structure-rich images for which our goal is to study how to preserve the structure. The diversity score should be interpreted jointly with the SIFID."

---

### Official Review · Reviewer_NUVi · 2022-07-08

**Rating:** 5
**Confidence:** 4
**Soundness:** 2 fair
**Presentation:** 2 fair
**Contribution:** 1 poor

**Summary:**

This work presents to extend light-weight SE module to be spatially-adaptive to improve model's efficiency and effectiveness. The proposed module is applied to both image synthesis tasks and image recognition tasks with a little different design. Its effectiveness is validated by  experiments on low-shot image synthesis and image classification to some extent.

**Questions:**

Please check that in weaknesses.

**Limitations:**

Authors mention the limitations and potential engative societal impact, however, they do not give detailed anaylsis and constructive suggestions.

**Strengths And Weaknesses:**

Strengths:
A lightweight SE module is applied to image synthesis tasks and image recognition tasks, whose effectiveness is validated by the corresponding experiments to some extent.

Weaknesses:
1. Novelty is quite limited, modules similar to the proposed spatially adaptive SE has been proposed before, such the ones in CBAM, RCAN and SFTGAN, SPADE. Please explain in detail what the main difference from these modules is.
2. The organization of this work look more like a technical report rather than an academic paper.

==========================post rebuttal==========================
Although authors addressed my comments to some extent, I still have a little concern about the novelty of this work. Therefore, I only raise the score from reject to borderline accept.

---

> ### Author Response · Authors · 2022-08-02
> **Response 1/1**
>
> Thank you for your time and efforts reviewing our paper.
>
> **Comments**: Novelty is quite limited, modules similar to the proposed spatially adaptive SE has been proposed before, such the ones in CBAM, RCAN and SFTGAN, SPADE. Please explain in detail what the main difference from these modules is.
>
> Response: Although integrating spatial and channel-wise feature attention schema has been extensively studied in the literature, the novelty of the proposed SASE lies in computing the 3D attention/re-calibration matrix (Fig.2) that re-couples layout-style decoupled information in end-to-end training. In the literature, there are many variants of the SE method, but the joint 3-D attention based variant in the proposed method is new. For example, the CBAM method (Woo et al, ECCV2018) learns the sequential integration of them. In SPADE, ground-truth semantic label maps are used in learning the spatially-adaptive feature normalization.  Learning the layout-style decoupling-coupling based 3D attention matrix in our SASE improves the capability of learning better feature exploration and exploitation in a backbone DNN for both generative and discriminative learning tasks, as the experimental results shown in both SASE-FastGANs, SASE-SinGANs and SASE-ResNet50.
>
> **Comments**: The organization of this work look more like a technical report rather than an academic paper.
>
> **Response**: It will be more helpful for us revising the submission if more specific suggestions were provided.
>
> **Comments**:  In addition to supplementary material, authors attach appendix in the main paper submission, which kinds of violates paper submission guideline.
>
> **Response**:  Before submission, we checked the guideline, and it seems that the appendix is allowed.

---

> > ### Comment · Reviewer_NUVi · 2022-08-08
> > **Further comments**
> >
> > 1. Authors do not explain well what the essential difference between the mentioned attention and modulation modules and the proposed one is. Why does the SASE module perform better than other modules? Authors are suggested to give corresponding explanation or experimental results for support.
> > 2. What I mean here is that explanation about the simple core module and the insight why it is advantageous over others is too short, while authors show many experiments about the application of the module (image synthesis and recognition). Authors are suggested to give more analysis about the proposed method and the insight about its effectiveness.
> > 3. I check it again. It is my fault and I am sorry for that.

---

> > > ### Author Response · Authors · 2022-08-08
> > > **Re: Further comments**
> > >
> > > Thank you very much for the further comments.
> > >
> > > **Comments**: Authors do not explain well what the essential difference between the mentioned attention and modulation modules and the proposed one is. Why does the SASE module perform better than other modules? Authors are suggested to give corresponding explanation or experimental results for support.
> > >
> > > > First of all, integrating spatial and channel-wise feature attention schema has been extensively studied in the literature. Most of them were variants of SE (for channel-wise attention) or/and Non-local networks (for spatial attention).  **How the spatial and channel-wise feature attention schema are actually integrated is the key for more effective and efficient learning.**
> > >
> > > > For the proposed SASE, the novelty lies in computing the 3D attention/re-calibration matrix (Fig.2) that *re-couples spatial-channel decoupled information* in end-to-end training, where the the spatial branch aims to model the object layout, and the channel branch aims to model the object appearance/style.
> > >
> > > > More specifically, the proposed SASE is motivated by the *split-transform-aggregate* design heuristic popularized by the well-known Google's Inception networks. Consider the SASE module for image synthesis, an input source feature map is first *split* into a number of groups (e.g., 4). Each group is then *transformed* to a latent style vector (via channel-wise attention) and a latent spatial mask (via spatial attention). The learned latent masks and latent style vectors are *aggregated* to form the full 3-D attention weights modulating the target feature map. The Inception design also induces fine-grained style learning and mixing between different groups split from a source feature map.  Consider the SASE module for image classification, it is specifically connected and compared with the Transformer model (Eqn.6 to Eqn.13) to justify its effectiveness.
> > >
> > > > For the attention methods mentioned by you,  they do not explore the joint re-calibration mechanism, but the sequential integration in CBAM and the ground-truth semantic label map driven SPADE, to name a few.
> > >
> > > > We do provide comprehensive experiments to show the effectiveness of the proposed SASE.
> > >
> > > **Comments**: What I mean here is that explanation about the simple core module and the insight why it is advantageous over others is too short, while authors show many experiments about the application of the module (image synthesis and recognition). Authors are suggested to give more analysis about the proposed method and the insight about its effectiveness.
> > >
> > > > Since the proposed SASE is indeed simple and intuitive neural architectural module, we think that the best way of justifying it should be the comprehensive experiments, which also refers to your comments above.  Due to the space limit, we did not manage to provide more explanations.  That being said, we will include more explanations in revision by moving some experiments into the appendix.

---

### Official Review · Reviewer_EeP1 · 2022-07-11

**Rating:** 3
**Confidence:** 4
**Soundness:** 1 poor
**Presentation:** 2 fair
**Contribution:** 1 poor

**Summary:**

This paper propose a self-attention module(SASE) for image synthesis and image recognition.
The self-attention is based on the **S**queeze-**E**xcitation(SE) mechanism, which is also
free from matrix multiplication.
By inplacing the matrix multiplication by Hadamard product and using the SE mechanism,
the computational efficiency improves while preserving decent performance.
Sufficient comparisons to traditional attention (not self-attention) mechanism demonstrate
the effectiveness of SASE in image synthesis.
While in image recognition, the SASE has slightly disadvantage compared with
Swin-tiny([32] in the main paper), but with only 67% parameters and 75% FLOPs.

**Questions:**

1. Why your module is effective in the few-shot and one-shot image systhesis? Is there any special design for this task?
   If so, what is it? (Q for weakness 1)
2. Why the SE mechanism is adopted in the $Q, K, V$ calculating procedure? Please explain your intention. (Q for weakness 3)
3. Why the authors choose $K$ (a.k.a spatial attention) for normalizing $A$ in figure 2,
   instead of choosing $Q$ (a.k.a channel attention) for normalizing $A$? (Q for weakness 3)
4. The traditional cross attention in transformer requires that the $K, V$ comes form the same feature[2][3] for better refining.
   However, the authors chooses $Q, K$ from the same feature. Please explain your intention. (Q for weakness 3)

[2] Chen, Chun-Fu Richard, Quanfu Fan, and Rameswar Panda. "Crossvit: Cross-attention multi-scale vision transformer for image classification." ICCV 2021.

[3] Vaswani, Ashish, et al. "Attention is all you need." NeruIPS 2017.

**Limitations:**

Yes, the authors adequately the limitations of their work.

**Strengths And Weaknesses:**

**Strengths**
1. The proposed SASE module is parameter-less and computationally inexpensive,
   while with respectable performance
2. The proposed SASE module is robust and easy to use. With simply replacing modules
   in existing network structures, performance can be improved.
3. The the operation of weighting $Q$ with $K$ in the SASE for image systhesis is novel,
   which is an effective way for aggregating the spatial attention and channel attention.

**Weakness**
1. This authors need to explain that why their module is effective in the few-shot and one-shot image systhesis.
   For example, in FastGAN([6] in the main paper), they proposed a self-supervised Discriminator in Auto-Encoder manner
   to fully utilize the information for the encoder in the discriminator to extract a more comprehensive representation.
   That's an effective strategy they designed individually for few-shot image systhesis.
2. There is no ablation studies in this paper, so that the design of the module is not guaranteed to be optimal.
   Such as where to adopt the SE mechanism is effective? In the $Q,K,V$ procudure or after the self-attention?
3. This paper, which does not explain some of the original intentions of the module design, can not convince me.
   Such as why the authors choose $X$ to compute the $Q, K$ and $Y$ for the $V$ instead of $X$ for the $Q$ and $Y$ for $K, V$?
   (in the SLE SASE module)
4. Using the self-attention mechanism, the authors need to compare their modules with currently *state-of-the-arts*
   self-attention modules in Table 3. Such as Swin-Transformer block([32] in the main paper) and Focal transformer Block[1] (at least one of them).

[1] Yang, Jianwei, et al. "Focal attention for long-range interactions in vision transformers." NeruIPS 2021.

---

> ### Author Response · Authors · 2022-08-02
> **Response 1/1**
>
> Thank you for your time and efforts reviewing our paper.
>
> **Comments**: Why your module is effective in the few-shot and one-shot image systhesis? Is there any special design for this task? If so, what is it?
>
> **Response**:  For few-shot image synthesis, we build on the Fast-GAN by only replacing their SLE module with the proposed SASE module. We use the same settings of the discriminator. Our goal is to provide an alternative to the SLE with better performance.  For single-shot image synthesis, one of the challenge of SinGANs is how to preserve structural information in synthesis. The spatially-adaptive aspect of the proposed SASE can significantly improve it.
>
> **Comments**:  There is no ablation studies in this paper, so that the design of the module is not guaranteed to be optimal. Such as where to adopt the SE mechanism is effective? In the Q, K, V procudure or after the self-attention?
>
> **Response**:  The design of the proposed SASE module is shown in Fig.2. It consists of spatial and channel-wise SE in parallel to achieve the 3-D attention mechanism. It adopts a different approach from the multi-head self-attention module used in the Transformer as shown in Eqn.6 to Eqn.13.
>
> **Comments**: This paper, which does not explain some of the original intentions of the module design, can not convince me. Such as why the authors choose X to compute the Q, K and Y for the V instead of  X for the Q and Y for K, V ? (in the SLE SASE module).
>
> **Response**:  When applying the propose SASE using the skip-layer settings in image synthesis, X and Y represent feature maps at different stages with different resolutions, and the goal is to learn a 3D attention matrix from X to recalibrate Y, where Q represents the channel-wise attention and K represents the spatial (gating) attention. It is computationally doable to use X to compute Q and Y for K, V, but then we will only able to leverage channel-wise attention from X, which is not our design intuition of learning spatially-adaptive SE. We will try this out in revision and compare it with the vanilla design.
>
> **Comments**: Why the authors choose K (a.k.a spatial attention) for normalizing  A in figure 2, instead of choosing Q (a.k.a channel attention) for normalizing A?
>
> **Response**:  The spatial attention K is trying to learn the semantic masks of objects and object parts, which are exclusive to each other in terms of the spatial occupancy, so we choose the spatial attention for normalizing A is to enable the network to be spatially adaptive and semantically aware.  If we use Q, we need to replicate it at very spatial location.
>
>
> **Comments**: The traditional cross attention in transformer requires that the K, V comes form the same feature[2][3] for better refining. However, the authors chooses Q, K from the same feature. Please explain your intention
>
> **Response**:  For the skip-layer setting used in image synthesis, as explained in the third comment above, Q, K are computing from the same feature to capture the channel and spatial attention.  For image classification (the right of Fig.2),  Q, K, V are computed from the same features.

---

### Official Review · Reviewer_f98y · 2022-07-12

**Rating:** 5
**Confidence:** 2
**Soundness:** 2 fair
**Presentation:** 2 fair
**Contribution:** 2 fair

**Summary:**

This work proposes to extend the widely adopted light-weight Squeeze-Excitation (SE) module to be spatially-adaptive to reinforce its data specificity, while retaining the efficiency of SE and the inductive basis of convolution. It presents two designs of spatially-adaptive squeeze-excitation (SASE) modules for image synthesis and image recognition respectively. For image synthesis tasks, the proposed SASE is tested in both low-shot and one-shot learning tasks and shows better performance than prior arts. For image recognition tasks, the proposed SASE is used as a drop-in replacement for convolution layers in ResNets and achieves better accuracy than the vanilla ResNets.

**Questions:**

See above.

**Limitations:**

Limitations and Potential Negative Impacts of the Proposed Work have been discussed in the paper.

**Strengths And Weaknesses:**

Strengths:
It presents a Spatially-Adaptive Squeeze-Excitation module with two realizations for better learning of generative models from low-shot / one-shot images.
It shows better performance for high-resolution image synthesis at the resolution of 1024×1024 when deployed in the FastGANs and better performance in image classification and object detection with smaller models.
It enables a simplified workflow for SinGANs, and shows a stronger capability of preserving image structures than prior arts.
The experiments are comprehensive.



Weaknesses:
1. The formulation introduction part is problematic. The current introduction includes too much detailed designs network structures. The introduction should more focus on the motivation and deliver the contribution in a high-level, instead of giving the detailed network design which should be appear in the method section.
2. Why put the related work section after the method section?
3. line 251, interms --> in terms

---

> ### Author Response · Authors · 2022-08-02
> **Response 1/1**
>
> Thank you for your time and efforts reviewing our paper.
>
> **Comments**:  The formulation introduction part is problematic. The current introduction includes too much detailed designs network structures. The introduction should more focus on the motivation and deliver the contribution in a high-level, instead of giving the detailed network design which should be appear in the method section.
>
> **Response**:  Thank you for the suggestion. We will carefully reorganize and revise the paper in revision.
>
>
> **Comments**:  Why put the related work section after the method section?
>
> **Response**:  We use this style following some of NeurIPS papers we read in the past. Our intent was trying to focus directly on the proposed method and then discuss its pros and cons in the context of relate work.

---

### Official Review · Reviewer_Xsyp · 2022-07-14

**Rating:** 4
**Confidence:** 3
**Ethics Flag:** Yes
**Soundness:** 3 good
**Presentation:** 1 poor
**Contribution:** 3 good

**Summary:**

Authors propose a Spatially-Adaptive Squeeze-Excitation drop-in module for GAN-base image synthesis and recognition, with improved accuracy and reduced computational cost.

**Questions:**

Generalization ability is unconvincing.

Visual effect of the results should be compared too.

**Ethics Review Area:**

["Inadequate Data and Algorithm Evaluation", "Inappropriate Potential Applications & Impact  (e.g., human rights concerns)"]

**Limitations:**

Not really. The module could potentially help synthesize or fake very realistic facial images and videos, hence may have negative social impact.

**Strengths And Weaknesses:**

Strengths: the proposed Spatially-Adaptive Squeeze-Excitation seems rational and meaningful

Weaknesses: limited novelty, weak presentation (typos, verbose statements, too small diagrams, ...)

---

> ### Author Response · Authors · 2022-08-02
> **Response**
>
> Thank you for your time and efforts reviewing our paper.
>
> **Comments**:  limited novelty
>
> **Response**: Although integrating spatial and channel-wise feature attention schema has been extensively studied in the literature, the novelty of the proposed SASE lies in computing the 3D attention/re-calibration matrix (Fig.2) that re-couples layout-style decoupled information in end-to-end training. In the literature, there are many variants of the SE method, for example, the scSE (Roy, Navab and Wachinger, MICCAI 2018) method does not ``re-couple" the concurrent spatial and channel SE information before feature re-calibration (Fig.1 (d) in the scSE paper), while the CBAM method (Woo et al, ECCV2018) learns the sequential integration of them.  Learning the layout-style decoupling-coupling based 3D attention matrix in our SASE improves the capability of learning better feature exploration and exploitation in a backbone DNN for both generative and discriminative learning tasks, as the experimental results shown in both SASE-FastGANs, SASE-SinGANs and SASE-ResNet50.
>
> For the few-shot image synthesis task, the proposed SASE outperforms the baseline CBAM and SPAP (Table.1 and 3). For the SinGAN task, the proposed SASE enables a substantially different pipeline (Fig.3 right). For the ImageNet classification task with ResNets, the proposed SASE enables more efficient yet more effective variants (Table 8).
>
> **Comments**:  weak presentation (typos, verbose statements, too small diagrams, ...)
>
> **Response**:  We will carefully revise and proofread the paper in revision.
>
> **Comments**: Generalization ability is unconvincing.
>
> **Response**: With all due respect, we do not know what you refer to specifically. In general, the proposed SASE module inherits the merits of the widely used SE module. We also show that it can be applied in three tasks: few-shot GAN, single-shot GAN and image classification.
>
> **Comments**: Visual effect of the results should be compared too.
>
> **Response**: We provided thorough qualitative results and comparisons in the Appendix (Fig. 7 to Fig. 25).

---

### Review · Ethics_Reviewer_iDwR · 2022-08-04

**Recommendation:**

The authors should avoid stating that there are no negative potential uses of this technology, and rather state how the applications that their methodology is enabling may lead to potential negative social impact. Note that acknowledging potential negative impacts and uses does not preclude the publication of the work.

**Ethical Issues:**

Yes

**Ethics Review:**

I do not understand why this paper was flagged for "Inadequate Data and Algorithm Evaluation", please note that this flag should only be used when the data and evaluation is inappropriate from an ethics perspective, and not when reviewers think that there are any types of flaws in evaluation.

I do understand why it was flagged for "Inappropriate Potential Applications & Impact". As the reviewer notes, synthesizing and faking realistic facial images and videos may have negative social impact. Image detection may also have negative effects, as exemplified by the increasing use many governments are giving it in the context of mass surveillance.

This does not mean that the work should not be published. But I believe that the statement "The proposed SASE module does not show
any potential negative impacts with its current form" is misleading. I invite the authors to reflect on the applications of image detection and image synthesis to provide a more informative response to the question "Did you discuss any potential negative societal impacts of your work?".

---

### Review · Ethics_Reviewer_Tdt8 · 2022-08-05

**Recommendation:**

Although these risks are real (and already observed in existing systems), they are not unique to this submission, affecting most generation and recognition models.  I would strongly encourage the authors to discuss these limitations of the current evaluation methodologies.


**Ethical Issues:**

Yes

**Ethics Review:**

The submission presents a new algorithm for image synthesis and recognition.

In terms of harms, image synthesis, especially of human faces, is a very real risk in terms of misrepresentation of specific individuals (e.g. deep fakes) or classes of people (e.g. astroturfing).  Since these uses of image generation are common, the authors’ improvement in performance is unlikely to amplify this problem too much.  However, it does legitimize a problematic domain.

Separate from this task-level harm, the authors do not examine harms related to stereotyping and other representational harms that can surface.  Whether and how performance varies across demographic groups is a substantial concern, especially when they can entrench existing inequities.  This is true of both generation and recognition tasks.

---

### Meta-Review · Area_Chair_ft8R · 2022-08-22

**Recommendation:** Reject
**Confidence:** Certain

**Metareview:**

This work proposes to extend the widely adopted light-weight Squeeze-Excitation (SE) module to be spatially-adaptive to reinforce its data specificity while retaining the efficiency of SE and the inductive basis of convolution. All reviewers have recognized the merit of the work, but raise critical concerns on presentation, insufficient justification of methods, and ethical issues. Specifically, the paper does not well explain why the proposed modules would necessarily perform well. This, however,  is important for a NeurIPS submission. Moreover, the risks of the method are not properly discussed while the paper has the statement like "The proposed SASE module does not show any potential negative impacts with its current form", which somehow is misleading. Last, the experiments may need to be carefully re-orgnized, and more visualization results would be helpful for understanding the proposed methods.

**Award:**

No

---

### Decision · Program_Chairs · 2022-09-14

Reject